# Analysis of the cloud fraction adjustment to aerosols and its dependence on meteorological controls using explainable machine learning

Yichen Jia [1, 2], Hendrik Andersen [1, 2], and Jan Cermak [1, 2]

[1]Karlsruhe Institute of Technology (KIT), Institute of Meteorology and Climate Research, Karlsruhe, Germany
[2]Karlsruhe Institute of Technology (KIT), Institute of Photogrammetry and Remote Sensing, Karlsruhe, Germany

**Correspondence:** Yichen Jia (yichen.jia@kit.edu)

**Abstract.** Aerosol-cloud interactions (ACI) have a pronounced influence on the Earth's radiation budget but continue to pose one of the most substantial uncertainties in the climate system. Marine boundary-layer clouds (MBLCs) are particularly important since they cover a large portion of the Earth's surface. One of the biggest challenges in quantifying ACI from observations lies in isolating adjustments of cloud fraction (CLF) to aerosol perturbations from the covariability and influence of the local meteorological conditions. In this study, this isolation is attempted using nine years (2011–2019) of near-global daily satellite cloud products in combination with reanalysis data of meteorological parameters. With cloud-droplet number concentration ($N_d$) as a proxy for aerosol, MBLC CLF is predicted by region-specific gradient boosting machine learning models. By means of SHapley Additive exPlanation (SHAP) regression values, CLF sensitivity to $N_d$ and meteorological factors as well as meteorological influences on the $N_d$–CLF sensitivity are quantified. The regional ML models are able to capture on average 45 % of the CLF variability. Based on our statistical approach, global patterns of CLF sensitivity suggest that CLF is positively associated with $N_d$, particularly in the stratocumulus-to-cumulus transition regions and the southern hemispheric midlatitudes. However, $N_d$ retrieval bias may contribute to non-causality in these positive sensitivities, and hence they should be considered as upper-bound estimates. CLF sensitivity to estimated inversion strength (EIS) is ubiquitously positive and strongest in tropical and subtropical regions topped by stratocumulus and within the midlatitudes. Globally, increased sea-surface temperature (SST) reduces CLF, particularly in stratocumulus regions. The spatial patterns of CLF sensitivity to horizontal wind components in the free troposphere may point to the impact of synoptic-scale weather systems and vertical wind shear on MBLCs. The $N_d$–CLF relationship is found to depend more on the selected thermodynamical variables than dynamical variables, and in particular on EIS and SST. In the midlatitudes, a stronger inversion is found to amplify the $N_d$–CLF relationship, while this is not observed in the stratocumulus regions. In the stratocumulus-to-cumulus transition regions, the $N_d$–CLF sensitivity is found to be amplified by higher SSTs, potentially pointing to $N_d$ more frequently delaying this transition in these conditions. The expected climatic changes of EIS and SST may thus influence future forcings from the CLF adjustment. The novel data-driven framework, whose limitations are also discussed, produces a quantification of the response of MBLC CLF to aerosols taking into account the covariations with meteorology.

## 1 Introduction

The emission of aerosols into the atmosphere affects the Earth's climate, in particular by masking part of the warming effect from greenhouse gases by reflecting solar radiation and changing cloud properties. Aerosol-cloud interactions (ACI) can strongly influence the Earth's energy distribution and thus also contribute a substantial uncertainty to past and future climate projections. The effective radiative forcing due to ACI (ERFaci) is assessed as $-1.0 \, \mathrm{Wm^{-2}}$ with an uncertainty range of $-1.7$ to $-0.3 \, \mathrm{Wm^{-2}}$ (Forster et al., 2021), albeit decades of effort and headway have been made in understanding the complex system of aerosols, clouds and their environmental controls. The correct representation of ACI in Earth system models (ESMs) remains a tremendous challenge because of the lack of accurate global quantification of the cloud-related fine-scale processes, and the lack of larger-scale constraints from the existing measurement systems at the ESM spatiotemporal resolution (Fan et al., 2016; Seinfeld et al., 2016; Sato et al., 2018).

Marine boundary layer clouds (MBLCs) cover over 23 % of the global ocean surface (Wood, 2012). Due to relatively small temperature differences between MBLC top and the sea surface, they only weakly impact outgoing longwave radiation but greatly reflect incoming shortwave radiation, leading to a strong net cooling effect (Hartmann et al., 1992). MBLCs play a critical role in the Earth's radiative balance (Zheng et al., 2021) and, in this regard, are the most important cloud type (Chen et al., 2014). Furthermore, MBLCs are especially susceptible to aerosol perturbations due to their relatively low optical depths (Turner, 2007; Leahy et al., 2012) and their formation in environments typically characterized by lower anthropogenic aerosol loading than continental clouds (Platnick and Twomey, 1994). Therefore, a deeper understanding of the aerosol-MBLC interactions is crucial to reduce the uncertainties in climate predictions. Atmospheric aerosols are critical for the formation of clouds as cloud condensation nuclei (CCN). Increases in aerosols are associated with increases in cloud droplet number concentration ($N_d$). As the cloud water is distributed among more droplets, cloud droplet effective radius ($r_e$) shrinks at constant liquid water content, resulting in an enhancement of cloud brightness and a negative instantaneous radiative forcing (Twomey, 1977). The likelihood of collision and coalescence subsequently decreases due to smaller drop sizes, hampering rainfall formation, which can prolong cloud lifetime and thus increase cloud fraction (CLF) (Albrecht, 1989). However, the aerosol–CLF relationship is complex, and the sign of the CLF adjustment can also be the opposite. This has been found in particular for non-precipitating clouds, stemming from enhanced entrainment mixing with ambient air over the clouds owing to shorter evaporation timescales (Wang et al., 2003; Jiang et al., 2006; Small et al., 2009) or reduced sedimentation (Ackerman et al., 2004; Bretherton et al., 2007) because of smaller droplet sizes.

From the perspective of observations at satellite scales, though there are studies suggesting a negative relationship between aerosols and CLF (Dey et al., 2011; Small et al., 2011), it has been documented by multiple studies that the overall CLF increases in response to increasing aerosols (e.g. Kaufman and Koren, 2006; Yuan et al., 2011; Gryspeerdt et al., 2016; Christensen et al., 2017; Andersen et al., 2017; Fuchs et al., 2018; Rosenfeld et al., 2019; Christensen et al., 2020). Likewise, studies based on ESMs reported substantial negative ERFaci due to liquid water path (LWP) and CLF adjustments (e.g. Zelinka et al.,

2014). In spite of the attribution of such adjustments in ESMs primarily to LWP adjustments (Ghan et al., 2016), a global satellite-based study by Bender et al. (2019) suggested that LWP adjustments are overestimated in ESMs, and that aerosol impact on CLF dominates the negative aerosol forcing. This is supported by observational evidence presented by Toll et al. (2019) who also reported an overestimation of LWP adjustment in climate models, and by Chen et al. (2022b) who recently highlighted the role of CLF increases due to aerosols from a large volcano eruption as the main cause of the associated forcing. Some large-eddy simulations have however suggested a negative response of CLF of trade wind cumulus to aerosol perturbations (Xue and Feingold, 2006; Seifert et al., 2015). While most studies, both from observational and model points of view, are in agreement that generally CLF increases with increasing aerosols due to a prolonged lifetime (Douglas and L'Ecuyer, 2022), the magnitude of the response of CLF to aerosols and its corresponding adjustments are still highly uncertain. For satellite-based analyses, one of the most challenging aspects in the quantification of CLF adjustment is isolating the influence of the aerosol loading on cloud properties from confounding covariations with meteorological parameters (Andersen et al., 2016; Gryspeerdt et al., 2019; Bellouin et al., 2020), paired with aerosol retrieval issues related to aerosol swelling and 3D radiative effects in the vicinity of clouds (Loeb and Schuster, 2008; Schwarz et al., 2017). Recent observational studies have utilized different methods to tackle this issue. A first approach is to stratify the data by meteorological factors and therefore accounting for local meteorology in the relationships (e.g. Su et al., 2010; Chen et al., 2014; Andersen and Cermak, 2015). Secondly, using $N_d$ as a mediating variable was proposed by Gryspeerdt et al. (2016) to analyze the causal pathway between aerosol optical depth and CLF. Another approach is to use a sampling strategy that applies a cloud–aerosol pairing algorithm (Christensen et al., 2017). However, these methods do not account for aerosol retrieval issues, meteorological influencing factors and confounders at once, which is essential to constrain the CLF adjustment. Recently, several studies have successfully used machine learning (ML) to account for non-linearities and meteorological factors to quantify ACI (Andersen et al., 2017; Fuchs et al., 2018; Dadashazar et al., 2021; Zipfel et al., 2022). ML regression algorithms allow to predict CLF (predictand) on the basis of aerosol and meteorological factors at the same time and treat the aerosol-cloud-meteorology system as a whole. In addition, ML models can represent non-linear interactive systems, which can be analyzed in sensitivity analyses with explainable ML techniques. Explainable ML refers to the techniques explaining the predictions of a trained ML model by explicitly quantifying the relationships, which helps improve the understandability, transparency and trustworthiness of the ML models (**?**).

In this study, we set up region-specific ML models at a global scale using satellite and reanalysis data sets to predict CLF to analyze $N_d$-induced changes in MBLCs. The goal of the explainable ML framework is to quantify the global sensitivity patterns of CLF to $N_d$ and meteorological factors. In addition, we aim to estimate the magnitude of the dependence of $N_d$–CLF sensitivity on the meteorological factors using SHapley Additive exPlanation (SHAP) interaction values, providing a new and insightful pathway to more profound knowledge of the physical processes relevant to the CLF adjustment and hence to a global constraint on aerosol-induced CLF changes accounting for meteorological covariations. The hypothesis of this study is that the response of cloud fraction of MBLCs to aerosol perturbations is positive, but buffered, i.e. reduced or amplified, by ambient meteorology and both the sensitivities and the interactions with meteorological factors have distinct regional patterns.

## 2  Data and methods

### 2.1  Data sets

This work combines nine years (2011-2019) of satellite retrievals from Moderate Resolution Imaging Spectroradiometer (MODIS) and reanalysis data from the European Centre for Medium-Range Weather Forecasts (ECMWF) from 60°N to 60°S. In this study, MBLCs are defined as single-layer warm cloud fields with cloud top temperatures higher than 268 K. To achieve this, the information on CLF (product: Cloud_Retrieval_Fraction_1L_Liquid), $r_e$ (product: Cloud_Effective_Radius _1L_Liquid_Mean), Cloud optical depth ($\tau_c$; product: Cloud_Optical_Thickness_1L_Liquid_Mean), cloud top temperature (CTT; product: Cloud_Top_Temperature_Mean) and satellite viewing geometry are obtained from MODIS level-3 collection-6.1 atmosphere daily products on the Terra platform (MOD08_D3), which are gridded into $1° \times 1°$ globally from level-2 atmospheric products. CLF serves as the predictand in this study. The computation of $N_d$ relies on $\tau_c$ and $r_e$, with filtering criteria based on CTT, solar zenith viewing angle and satellite zenith angle, as elaborated in the following.

The equation used to calculate the MODIS $N_d$ is from Quaas et al. (2006), which depends on the retrievals of $r_e$ and $\tau_c$, so do the uncertainties on the errors propagated from $r_e$ and $\tau_c$:

$$N_d = \alpha \tau_c^{0.5} r_e^{-2.5} \tag{1}$$

where $\alpha = 1.37 \times 10^{-5}$ m$^{-0.5}$ is a constant related to adiabatic growth rate. The uncertainties in $N_d$ retrievals are exhaustively evaluated by (Grosvenor et al., 2018), which suggests that the uncertainties in averaged $N_d$ over $1° \times 1°$ grid box (spatial resolution of the MODIS products used in this study) decrease by over 50 % compared to pixel-level uncertainties. This derivation approach relies on the assumed adiabaticity in global marine warm clouds where liquid water content and $r_e$ increase monotonically and $N_d$ distributes as constant vertically. Departure from the adiabatic assumption (e.g. due to entrainment) would result in $N_d$ retrieval biases (Merk et al., 2016; Bennartz and Rausch, 2017). The uncertainty related to the estimation of $N_d$ from MODIS also depends on liquid CLF. $N_d$ is less biased in the regions of larger CLF where clouds are more homogeneous, while in the regions with lower CLF $N_d$ retrievals are sparser and less reliable (Grosvenor et al., 2018; Zhu et al., 2018). In such heterogeneous cloud fields, subpixel effects in the retrieval of $r_e$ can negatively bias the retrieved $N_d$ values (Zhang and Platnick, 2011; Zhang et al., 2012; Grosvenor et al., 2018). Such retrieval biases could cause a bias in the $N_d$–CLF relationship as well. Furthermore, the interpretation of the causal effect of $N_d$ on CLF can also be obscured by small-scale sampling issues. In particular, apart from the retrieval errors in $r_e$ and $\tau_c$, the natural spatial variability in cloud fields can also propagate to $N_d$ estimate and distort the $N_d$–CLF relationship (Arola et al., 2022; Liu et al., 2024).

Following the screening criteria for more reliable $N_d$ demarcated by Gryspeerdt et al. (2022), only clouds restricted to single-layer in liquid phase with CTT higher than 268 K are considered. As suggested by Quaas et al. (2006), samples with $r_e$ < 4 μm and $\tau_c$ < 4 are excluded to cope with the high $r_e$ retrieval uncertainties at low $\tau_c$. In addition, solar and sensor viewing zenith angles respectively greater than 65° and 55° are removed to avoid the large biases in $r_e$ and $\tau_c$ retrievals (as in Grosvenor et al., 2018). The pixels selected according to the above sampling strategies generate more reliable $N_d$ estimates.

Atmospheric and oceanic variables are taken from the fifth generation ECMWF atmospheric reanalysis of the global climate (ERA5) at an hourly frequency (Table 1) (Hersbach et al., 2020). The ERA5 data sets are harmonized to fit the level-3 MODIS data by first being resampled to $1° \times 1°$ from their default $0.25° \times 0.25°$ spatial resolution using bilinear interpolation, and they are subsequently collocated to Terra MODIS by extracting hourly data to align with the UTC overpass times of the Terra satellite for each grid cell, yielding a spatiotemporally matched MODIS-ERA5 combined data set for training the ML models. For $N_d$ retrievals, only samples within 1–99 percentiles are retained to exclude potential unrealistic outliers from $r_e$ and $\tau_c$ retrievals (Zipfel et al., 2022). Furthermore, the explanation of ML models in this study relies on using linear regressions to capture the distribution of individual prediction instances, and the extreme values may excessively magnify or reduce the sensitivity or interactive effects quantified by SHAP (shown in Fig. 1 and discussed in Sect. 2.3.2). The threshold of 1–99 percentiles for each predictor is thus adopted to remove the values at the very tails of the specific distribution and to improve the robustness of the estimated sensitivities. To define the sensitivities of CLF and the interactive effects of meteorological factors, the natural logarithm of $N_d$ is taken (see Sect. 2.3.2 in detail). Estimated inversion strength (EIS) is calculated based on the formulation from Wood and Bretherton (2006) and in this study, it is dependent only on atmospheric temperatures at 700 hPa and at the level of 1000 hPa.

All input predictors for each XGB model (i.e. for each $5° \times 5°$ window aggregated from $1° \times 1°$ grid boxes, as detailed in Sect. 2.2) are standardized by centering around the mean and scaling to have unit variance as in Scott et al. (2020). Hamby (1994) suggested that the standardization process is a standard practice when aiming for comparability of sensitivity estimates across predictors. This process eliminates the influence of units and aligns data on the same scale instead of the original natural ones, thereby ensuring comparability of the quantified sensitivities and interactive effects with meteorology among different variables. This standardization procedure has been applied in other studies investigating different cloud sensitivities to various cloud-controlling factors (e.g. Ceppi and Nowack, 2021; Andersen et al., 2023). This procedure however may result in reduced spatial comparability due to variations in mean and standard deviation values across different $5° \times 5°$ windows. To assess the trade-off between comparability among different predictors and comparability in space, we provide results without standardization in the supplementary material (Fig. S2 to Fig. S7 therein) as done by Grise and Kelleher (2021). In terms of spatial patterns, the results are nearly identical to their corresponding ones presented in the following sections of the main text, suggesting that standardizing the data based on the local mean and standard deviation for each window has only a small impact on comparability across each window. Therefore, we primarily benefit from achieving comparability among different predictors while only a minor compromise in spatial comparability.

## 2.2 Machine learning model setup

Extreme Gradient Boosting (XGB) is a distributed tree boosting algorithm aiming to provide a scalable, portable and flexible library under the Gradient Boosting framework (Chen and Guestrin, 2016). The state-of-the-art XGB can be implemented efficiently in Python and has been recently used to study clouds and ACI (Andersen et al., 2022; Douglas and L'Ecuyer, 2022). As an extension of previous gradient boosting methods, XGB has incorporated regularization techniques which help prevent overfitting and improve model generalization. Besides, the subsampling on training subsets and column (feature) subsampling

**Table 1.** Summary of the predictors from ERA5 reanalysis.

| Predictor Name | Abbreviation | Units |
|---|---|---|
| **Instantaneous pressure level parameters (at 700 hPa, 850 hPa)** | | |
| Relative humidity | $RH_{700}, RH_{850}$ | % |
| Specific humidity | $SH_{700}, SH_{850}$ | $kgkg^{-1}$ |
| Temperature | $t_{700}, t_{850}$ | K |
| Vertical velocity | $\omega_{700}, \omega_{850}$ | $Pas^{-1}$ |
| Eastward wind component | $u_{700}, u_{850}$ | $ms^{-1}$ |
| Northward wind component | $v_{700}, v_{850}$ | $ms^{-1}$ |
| **Surface and single level parameters (instantaneous or mean rates/fluxes)** | | |
| Eastward and northward wind component at 10 m | $u_{10}, v_{10}$ | $ms^{-1}$ |
| Boundary-layer height | BLH | m |
| Convective available potential energy | CAPE | $Jkg^{-1}$ |
| Sea surface temperature | SST | K |
| Total column water vapour | TCWV | $kgm^{-2}$ |
| Mean large-scale precipitation fraction | PF | Proportion |
| Mean surface sensible/latent heat flux | SHF/LHF | $Wm^{-2}$ |
| **Calculated** | | |
| Estimated inversion strength | EIS | K |

techniques can shorten the running time and also avert overfitting and hence elevate model performance (Chen and Guestrin, 2016). Relevant regularization and subsampling hyperparameters are tuned using Bayesian Optimization to determine the best combination, see Table 2 for the search space.

Data from 2011 to 2016 are used for training and data from 2017 to 2019 for testing (independent train/test split about 67 %/33 %). By chronologically splitting the training and test sets without random shuffling, we ensure that the training data will not see future information and the autocorrelation in data will not lead to overopstimic evaluation of the model's performance Beucler et al. (2023); Kapoor et al. (2023). As suggested by Karpatne et al. (2017), a single ML model may not perform well across all regions due to the heterogeneity of relevant processes. Therefore, data at a $1° \times 1°$ spatial resolution are aggregated into $5° \times 5°$ geographical windows, where an individual, independent XGB model is trained and tested for each "window". Hereby a region-specific ML framework is established to potentially capture regional relationships and characteristics and thus the regional patterns of CLF adjustment. The coarser $5° \times 5°$ spatial resolution of the modelling grid increases the sample size by a factor of $\approx 25$ which is helpful to establish robust sensitivity estimates. In addition, at the spatial resolution of

**Table 2.** Overview of the hyperparameters tuned for regional Extreme Gradient Boosting models using Bayesian Optimization.

| Hyperparameter name | Search Space |
| --- | --- |
| learning_rate | 0.01–0.5 |
| max_depth | 3–10 |
| min_child_weight | 1–10 |
| subsample | 0.5–1 |
| colsample_bytree | 0.5–1 |
| gamma | 0–10 |
| alpha | 0–10 |
| lambda | 0–10 |

$1° \times 1°$ summarized in $5° \times 5°$ degree windows, the spatial scale is adequate for ACI sensitivity estimation (Grandey and Stier, 2010). To ensure a sufficient data amount for training and testing the XGB models, only the geographical windows with over 6000 available data points are retained. Consequently, 34 out of 1190 oceanic windows have been excluded. These windows located between 47.5°W–122.5°E and 52.5°S–57.5°S in the Southern Ocean (Fig. 2) contain fewer than 6000 valid samples due to the screening for $N_d$ retrievals. For each model, the hyperparameters are tuned by implementing Bayesian optimization, which uses a Gaussian process prior distribution over hyperparameters to initialize a probabilistic model for the objective function to be optimized. After the initialization, the probabilistic model is updated iteratively and Bayesian optimization suggests the optimal combination of hyperparameters to try for the next iteration according to the previous one and samples gathered from the search space (Table 2) (Snoek et al., 2012). Each iteration is evaluated by 5-fold cross-validation using root mean square error (RMSE) as score. The number of boosting rounds (the number of trees) for each XGB model is then determined by the early stopping technique to further avoid overfitting i.e. the training of the model will stop early once it is monitored that the score of cross-validation does not improve within 20 iteration rounds.

## 2.3 Explaining the machine learning models

### 2.3.1 SHapley Additive exPlanation (SHAP) values

SHAP values were proposed by Lundberg and Lee (2017) on the basis of cooperative game theory to explain the outputs of ML models. The SHAP approach has been implemented with XGB in Python and it has been reported that outputs from XGB models with various number of trees can be well explained by the SHAP framework in different subject areas (e.g. Padarian et al., 2020; Lundberg et al., 2018, 2020; Kim et al., 2021; Li et al., 2022). The contribution of a predictor value to a specific model prediction is calculated as the difference between the predictions of the model in the presence and absence of this particular predictor for all possible combinations of predictor values. Since this is performed at a "local" level (i.e. for this

specific instance's prediction), it allows for insights into how a certain model outcome is achieved, thereby complementing more traditional "global" (considering all instances) feature importance measures (e.g. partial dependence plot).

The base value in the context of SHAP values is what would be predicted in the absence of any feature information (Lundberg and Lee, 2017) and it is typically computed as the average of all predictions by ML models over the entire training data set. Positive (negative) SHAP values indicate that the specific feature value increases (decreases) the prediction compared to this base value. In other words, the base value serves as the reference point against which the contributions of individual features are measured. SHAP values for all features will always sum up to the difference between the base value and the final model prediction so that SHAP values are additive and internally consistent. The base value could be analogous to the climatological CLF for a given geographical window assuming no information about the input parameters is known. In this context, the SHAP values of input features indicate the extent to which knowing information about each feature value would deviate the prediction from the climatological CLF (base value).

Furthermore, the quantification of the influence of meteorology on the $N_d$–CLF relationship can be analysed using SHAP interaction values, which are an extension of SHAP values. They measure the difference between the SHAP values for a feature when another (secondary) feature is included versus when it is not included, offering a potential tool for insights into feature interactions captured by the tree ensembles. SHAP values have been applied to study atmospheric aerosols in the context of air pollution Stirnberg et al. (2021), and have been used by Zipfel et al. (2022) to explore satellite-observed $N_d$-LWP relationship in MBLCs in the Southeast Atlantic, finding that meteorological variables have considerable influences on the $N_d$-LWP relationship using SHAP interactive values. Moreover, the use of SHAP interaction values in these studies allows for a more profound and in-depth comprehension of the underlying processes with respect to local meteorology. SHAP values provide insights into the behaviour of the XGB models, and as all statistical/ML models, they may not necessarily reflect real-world physical causality. Nevertheless, this state-of-the-art technique allows us to account for meteorological covariations when deriving sensitivities and to appraise to what extent the meteorological predictors interact with and influence the $N_d$–CLF relationship beyond traditional global-level feature attributions.

### 2.3.2 Quantification of sensitivities and interactive effects

Figure. 1 is an exemplary graph for a regional XGB model at a specific $5° \times 5°$ window ($27.5°$S–$32.5°$S, $122.5°$W–$127.5°$W). SHAP values and SHAP interaction values are used to explain this XGB model and to quantify and isolate the CLF sensitivity to $N_d$ and the interactive effects of meteorological factors (here sea-surface temperature (SST)). Each dot in Fig. 1 represents an individual data instance (i.e. a single observation at a specific grid cell and time step) and shows how individual $N_d$ or $\ln N_d$ values impact the CLF prediction.

Plotting SHAP values of $N_d$ against $N_d$ values without the standardization process (Figure 1 (a)) for each data sample illustrates that increased $N_d$ values lead to an increase in the predicted CLF, while the rate of the increase ($dSHAP/dN_d$) drops with $N_d$ as shown by the orange line. For each $20 \, \mathrm{cm}^{-3}$ wide bin of $N_d$, $dSHAP/dN_d$ is calculated as the slope of the linear regression between $N_d$ and $N_d$ SHAP values. The nonlinear positive association between $N_d$ and predicted CLF aligns well with findings of prior studies (e.g. Gryspeerdt et al., 2016; Rosenfeld et al., 2019) that the aerosol impact on CLF saturates

at relatively high aerosol loading. This relationship also resembles the one reported by Yuan et al. (2023) which is attributed to the precipitation suppression effect due to relatively high $N_d$.

Expressing the sensitivity logarithmically in $N_d$ is ideal because cloud processes are prone to respond to a relative change in $N_d$ rather than an absolute one (Carslaw et al., 2013; Bellouin et al., 2020). Furthermore, the log-transformed $N_d$ facilitates the application of simple linear regressions to capture the relationship between the contribution of $N_d$ to the predicted CLF ($N_d$ SHAP values) and its feature values. As depicted in Fig. 1 (b), the contribution of $\ln N_d$ to the predicted CLF increases almost linearly with rising $\ln N_d$. Thus, the CLF sensitivity to $N_d$ is estimated as the slope of the linear regression between $\ln N_d$ SHAP values and $\ln N_d$ values (0.098 CLF $\sigma^{-1}$). A similar method to estimate sensitivity has also been used by Li et al. (2022), where it is also suggested that this method can enhance the robustness of the sensitivity estimation. Because it can leverage the benefits of an XGB model, including bagging techniques and no need for distribution assumptions, along with the advantages of SHAP, which provides global interpretations consistent with local explanations (Lundberg et al., 2020; Molnar, 2022)). It should be noted that the notably linear relationship in Fig. 1 (b) does not hold across all geographical windows. Fig. S1 displays additional exemplary windows where the relationships exhibit less linearity. Our approach also captures nonlinearity in the system; in these cases, the linear regression helps decrease the convolved relationships as in Gryspeerdt et al. (2016). Note that unlike $N_d$ (cm$^{-3}$) in (a), $\ln N_d$ and SST in (b) and (c) have been standardized and thus sensitivities and IAIs are expressed with the unit of cloud fraction change per standard deviation (CLF $\sigma^{-1}$). Standardizing all predictors ensures that the results become comparable across all of them. We also present the SHAP dependence plots for the same example window in Fig. S2 where non-standardized $\ln N_d$ and SST are used to plot (b) and (c). The patterns are alike and only the magnitudes of the example sensitivity and IAI are different because they are no longer expressed on a physical scale.

The vertical dispersion around the $\ln N_d$–CLF relationship captured by the SHAP dependence plot is due to the dependence of the $\ln N_d$ contribution to the predicted CLF on meteorological factors (e.g. SST) in the model, which is captured by SHAP interaction values, as displayed in Fig. 1 (c). The colouring of the data points by SST illustrates how interactions with SST split up the $\ln N_d$–CLF relationship, with low SST values amplifying the $\ln N_d$ contribution and vice versa. To quantify this interaction effect, the meteorological data are then divided into a group of above-average feature values and a group of below-average feature values. A linear regression is fit to the $\ln N_d$ values and the SHAP interaction values in each group. An interaction index (IAI) is derived from these regression fits and defined as the slope for the high-value group (> mean) minus the slope for the low-value group (< mean):

$$IAI = \beta_{x,high} - \beta_{x,low} \tag{2}$$

where $\beta$ is the slope of the linear regression between SHAP interaction values and $\ln N_d$ values, and the subscripts denote the high-value group and the low-value group for a specific meteorological variable $x$ (SST in the example), respectively. At the exemplary geographical window, the influence of SST on the $N_d$–CLF sensitivity is quantified by IAI = -0.029 CLF $\sigma^{-1}$ (Fig. 1 (c)). Similar to sensitivities, the unit of IAIs is also CLF $\sigma^{-1}$. Therefore, for a positive sensitivity such as the $N_d$–CLF sensitivity shown in Fig. 1 (b), a negative IAI value means that the $N_d$–CLF sensitivity is larger with low feature values,

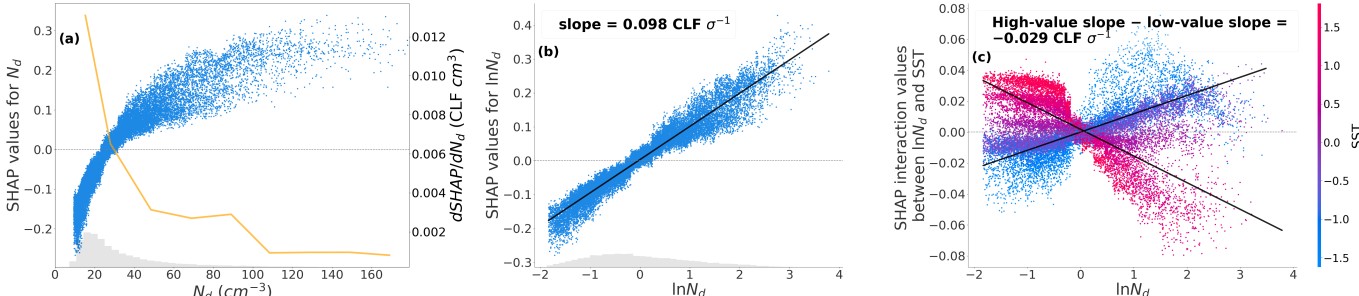

**Figure 1.** SHAP dependence plots for cloud droplet number concentration ($N_d$) in the region from 27.5°S to 32.5°S and from 122.5°W to 127.5°W. (a) Dots show $N_d$ SHAP values versus $N_d$ values. The orange line shows the change rate of $N_d$ SHAP values with respect to $N_d$ ($dSHAP/dN_d$) versus $N_d$ values for each $N_d$ bin of 20 cm$^{-3}$ wide. (b) similar to (a) but showing the relationship between $\ln N_d$ SHAP values and $\ln N_d$ with the corresponding sensitivity defined as the slope of the linear regression. (c) SHAP interaction values coloured by sea surface temperature (SST) showing the dependence of $\ln N_d$–CLF relationship on the interactive effects of SST. The interaction values are further divided into two groups by the mean feature value of SST. Linear regressions are performed respectively for the high-value group and low-value group and the Interaction Index (IAI) is defined as the slope for the high-value group subtracting the slope for the low-value group. The horizontal dashed lines are a demarcation between negative and positive SHAP (interaction) values. Note that $N_d$ in (a) is not standardized while $\ln N_d$ and SST in (b) and (c) are standardized.

as shown in Fig. 1 (c) (the positive relationship is weakened by high SST values). On the contrary, a positive IAI value is corresponding to a larger positive sensitivity with high feature values.

### 2.3.3 Limitations of observation-based machine-learning of aerosol-cloud processes

In this section limitations of this study are discussed. A fundamental limitation of our study is that the assertion of causality from the statistical relationships of aerosols/$N_d$ and cloud fraction/properties is not easily done. While causal inference approaches exist and have been applied in the field of aerosol-cloud interactions (Fons et al., 2023), we employ a more traditional approach of analyzing statistical relationships of instantaneous observations (i.e. correlations). Unless nonetheless explicitly incorporating such causal inference approaches, studies utilizing statistical or ML models to explore observational aerosol-cloud processes contend with this common limitation. For instance, some studies assessed satellite-based statistical relationships between CLF and $N_d$ (Christensen et al., 2016, 2017), between LWP and $N_d$ (Michibata et al., 2016; Rosenfeld et al., 2019), and between $N_d$ and other aerosol proxies (Gryspeerdt et al., 2017; McCoy et al., 2017a), all resting on statistically inferring sensitivities of cloud quantities to aerosol proxies (Forster et al., 2021). While we interpret the derived relationships with respect to the known physical relationships, uncertainties regarding the physical interpretation are mainly driven by two sources: uncertainties in the data and uncertainties from the methods.

1. Data: uncertainties exist for each satellite/reanalysis quantity, but may be particularly large in $N_d$. For example, the subpixel effect can introduce more bias in the $N_d$ retrieval process within broken-cloud regimes due to increased hetero-

geneity. The $N_d$ retrieval biases are discussed in Sect. 2.1. Also, $N_d$ and CLF observations are not fully independent, which may introduce a spurious positive correlation between the two variables. As such, we expect the physical relationship of $N_d$ and CLF to be weaker than our estimate, so that the derived sensitivities present an upper bound of the physical relationship.

Another caveat in our data is that $N_d$ values in our study are computed using MODIS level-3 large-scale mean $r_e$ and $\tau_c$ values instead of joint histograms as in Gryspeerdt et al. (2016). This may introduce additional biases considering the nonlinearity of the $N_d$ calculation. In future work, $N_d$ data calculated from underlying joint histograms or pre-filtered data by Gryspeerdt et al. (2022) could be applied to be compared with the results in this study.

2. Methods:

   a. The exact quantification of sensitivities is dependent on the choice of the statistical/machine learning model. While for (more linearly related) monthly data, Andersen et al. (2022) have shown that XGB, artificial neural networks and linear models tend to lead to very similar results, this is not expected for more instantaneous data. Here, nonlinear relationships are expected, and a more complex nonlinear model is a more appropriate choice. XGB and other tree ensemble methods are a particularly popular choice, because of their interpretability, high accuracy considering computational efficiency (Lundberg et al., 2020) and their ability to model the interactions between predictors (Elith et al., 2008). They have been used frequently to study aerosols and clouds before (Fuchs et al., 2018; Dadashazar et al., 2021; Andersen et al., 2021; Chen et al., 2022b; Bender et al., 2024). Besides, the Tree-SHAP algorithm, specifically tailored for tree-based models to compute exact SHAPley values, can even further enhance their interpretability and has been applied in this field as well (Stirnberg et al., 2021; Zipfel et al., 2022).

   b. The quantification of sensitivities with SHAP values depends on details: the choice of the algorithm to effectively estimate Shapley values is application-specific and comes to the trade-off between being *true to the data* and *true to the model*, which relies on an observational and interventional conditional expectation, respectively (Chen et al., 2020). The *true to the model* approach is preferable when trying to understand how an ML model makes a prediction, which requires assuming feature independence. In this study, we focus on potential mechanisms behind CLF sensitivities and thus we tend to respect the correlations spread among input features (*true to the data*) (Frye et al., 2021; Chen et al., 2022a). Consequently, we suffer from the disadvantage of being *true to the data*: entangled importance attributions of correlated features e.g. a feature not explicitly used by the model for the prediction task might be assigned a non-zero contribution. Yet we refrain from the drawback of being *true to the model*— unrealistic input instances (Sundararajan and Najmi, 2020; Linardatos et al., 2021; Chen et al., 2023). Despite the inherent trade-off, SHAP approach has been employed in the context of being *true to the data* (e.g. Stirnberg et al., 2021; Zipfel et al., 2022; Li et al., 2022).

The derived estimates of sensitivities and interactive effects in this paper should thus be interpreted with these limitations and uncertainties in mind.

# 3 Results and discussion

## 3.1 Model performance

The skills of the region-specific XGB models in predicting CLF are evaluated by the coefficient of determination ($R^2$) on the unseen hold-out test data. The global weighted mean $R^2$ is 0.45 (about 45 % on weighted average and up to 73.57 % of the variability in CLF prediction is explained) and a standard deviation of 0.10. While this means that on average, about half of the variability in CLF cannot be explained by the machine learning models, this is expected as previous studies have shown that the performance of statistical models decreases when going from monthly to daily data (Andersen et al., 2017; Fuchs et al., 2018; Dadashazar et al., 2021), and the performance is on par with that reported by Dadashazar et al. (2021), who used machine learning models to predict $N_d$ with daily reanalysis data. The models in tropical regions in the Indian Ocean and the Western Pacific relatively poorly explain the variability in CLF, while XGB models perform well in the stratocumulus regions in the subtropics near the continents, and in the midlatitudes, particularly the southern hemispheric midlatitudes. The high skill of predicting CLF in the southern hemispheric midlatitudes is in contrast to a recent study where this region has been found to be particularly difficult to model statistically with monthly data (Andersen et al., 2023). In this region, the day-to-day CLF variability is high due to the large influence of synoptic-scale weather systems, and hence data at the daily resolution is more adequate to represent the CLF variability in these regions.

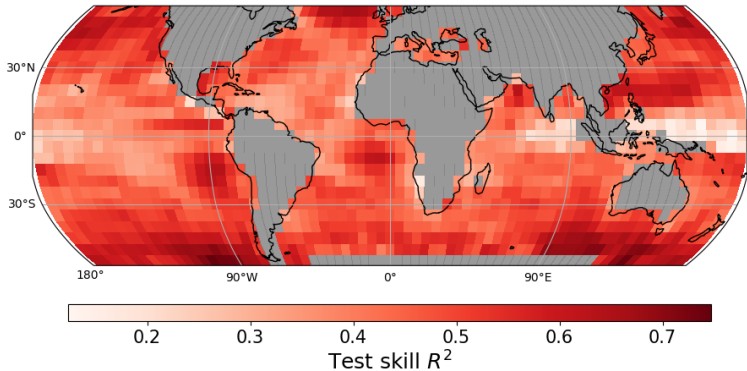

**Figure 2.** $R^2$ score of regional Extreme Gradient Boosting models predicting cloud fraction of marine boundary layer clouds in the independent test data set (2017–2019).

## 3.2 CLF sensitivity: global perspectives and regional characteristics

### 3.2.1 Global overview of CLF sensitivities

Figure 3 summarises the means and distributions of the near-global sensitivities of CLF to all predictors. The sensitivities are estimated as described in Sect. 2.3.2. The sequence is sorted by descending mean values of the absolute sensitivities (i.e.

by feature importance) of the predictor variables. A strong and consistently positive $N_d$–CLF sensitivity is found. The fact that CLF is most sensitive to $N_d$ is to be expected, as cloud observations from the same sensor are more directly related than a reanalysis product, so that their overall magnitude should not be compared (Zipfel et al., 2022). The entrainment of relatively dry air from the free troposphere into the MBL will be impeded by a stronger inversion (i.e. higher EIS), resulting in a shallower, better-mixed and more humid MBL conducive to stratocumulus clouds (Bretherton and Wyant, 1997; Wood and Hartmann, 2006; Qu et al., 2015a; Myers et al., 2021). The salient positive sensitivity to EIS is in accordance with the links found in previous studies (e.g. Klein and Hartmann, 1993; Qu et al., 2015b; Andersen et al., 2017) suggesting that EIS is a crucial controlling factor for low-marine cloud cover. Note that in some studies the strength of the inversion over the boundary layer is measured by lower tropospheric stability, which can be regarded as a similar metric outperformed by EIS (Wood and Bretherton, 2006). Precipitation fraction is the fraction of the original ERA5 grid box covered by large-scale precipitation. The strong positive CLF sensitivity to precipitation fraction is likely caused by the ML model learning that precipitation can be viewed as a proxy for cloudiness, rather than being an indicator of the physical processes via which precipitation exerts controls on the macrophysics of MBLCs. Humidity shows positive CLF sensitivities greater at 850 hPa, where cloud tops are often located (Gryspeerdt and Stier, 2012), than at 700 hPa which is typically in the free troposphere above the MBLCs (Myers and Norris, 2013). Likewise, the atmospheric temperature at 850 hPa ($t_{850}$) presents stronger CLF sensitivity than the temperature at 700 hPa ($t_{700}$). Nonetheless, in the case of winds the 700 hPa pressure level is more relevant than that at 850 hPa. A relatively pronounced negative sensitivity to the eastward wind component at 700 hPa ($u_{700}$) seems to indicate that clouds are depleted due to more westerlies at this level. CLF exhibits negative sensitivities to vertical pressure velocities both at 850 and 700 hPa, showing that large-scale ascending motion is connected to increases in MBLCs (Myers and Norris, 2013; Bretherton et al., 2013; Blossey et al., 2013). In general, the global averages of CLF sensitivity in terms of dynamical predictors (i.e. 3-D winds at surface and pressure levels) vary in sign and are less strong. A marked negative sensitivity of CLF to SST is found, which is in agreement to many prior studies (e.g. Qu et al., 2015b; Scott et al., 2020), where increases in SST have been found to lead to low cloud breakup and dissipation due to a number of processes as described in e.g. Scott et al. (2020). One of these is that the associated enhancement of mean surface latent heat flux (LHF) deepens MBL and facilitates buoyancy and thus the entrainment of dry free-tropospheric air (Rieck et al., 2012; Andersen et al., 2022). However, CLF is much less sensitive to LHF than to SST, which may indicate that this mechanism is less important at the spatial and time scales considered in this study. CLF exhibits a considerable negative sensitivity to mean surface sensible heat flux (SHF), which quantifies an increase in CLF with increasing SHF (upward SHF are negative). While increased SHF can promote the transition from decks of stratus or stratocumulus clouds (high CLF) to more convective clouds (low CLF) due to the deepening of the boundary layer (Fan et al., 2016), potentially leading to a positive SHF–CLF relationship, increased SHF is associated to situations of cold air advection where turbulent surface fluxes are enhanced, which could lead to marked increases in CLF (Miyamoto et al., 2018; Zelinka et al., 2018; Grise and Kelleher, 2021)

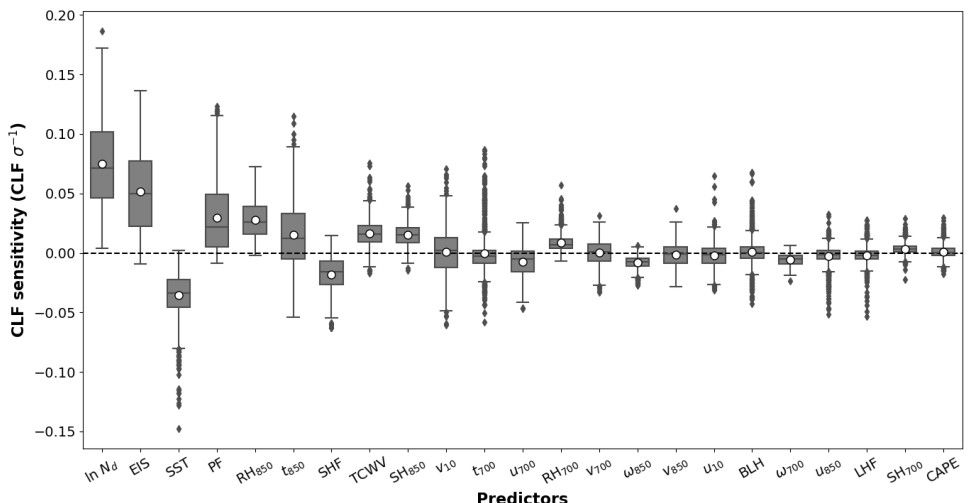

**Figure 3.** The distribution of the sensitivities of cloud fraction to all predictors as depicted in Table 1. Boxes represent the interquartile range which is extended by whiskers up to 1.5 interquartile ranges, with outliers shown as the points outside the range. The solid line and white dot in each box show the median and mean values of the sensitivities, respectively. Predictors are sorted by the mean values of absolute sensitivity values. The dashed line across the figure differentiates positive and negative sensitivity values.

### 3.2.2 Spatial patterns of the CLF sensitivity to $N_d$

The sensitivity of the MBLC fraction associated with the aerosol proxy $N_d$ are ubiquitously positive, in accordance with the global correlations or sensitivities found in (e.g. Gryspeerdt et al., 2016; Andersen et al., 2017). This is presumably due to the lifetime effect, but could also partially result from $N_d$ retrieval biases discussed in Sect. 2.1. The global weighted mean value of the $N_d$–CLF sensitivity is 0.074 CLF $\sigma^{-1}$ with a standard deviation of 0.036 CLF $\sigma^{-1}$. The relationship between CLF and $N_d$ is found particularly strong in the regions of frequent stratocumulus to cumulus transition off the western continental coasts. These marked positive $N_d$–CLF sensitivities may be caused by high $N_d$ delaying the transition from stratocumulus to cumulus clouds (Gryspeerdt et al., 2016; Christensen et al., 2020). However, as this cloud regime transition involves clouds shifting from more overcast to more broken, the strong relationships in these regions may be more affected by $N_d$ retrieval errors. The $N_d$–CLF sensitivity is also pronounced in the southern hemispheric midlatitudes, where stratiform clouds dominate. The $N_d$–CLF sensitivity is weak and close to 0 in the tropics, in particular in the deep convective warm pool region. These spatial patterns of $N_d$–CLF sensitivity resemble those found by Gryspeerdt et al. (2016), in particular the ones where they mediated the aerosol optical depth–CLF relationship by $N_d$, but are more pronounced in the southern hemispheric midlatitudes. This difference in estimated sensitivity seems noteworthy and should thus be investigated in future work. As $N_d$ retrievals tend to negatively bias at lower CLF and positively bias at higher CLF, the $N_d$–CLF sensitivity may be overestimated, and at the scales considered here, should be interpreted as an upper bound to the physical $N_d$–CLF sensitivity. The global weighted average of

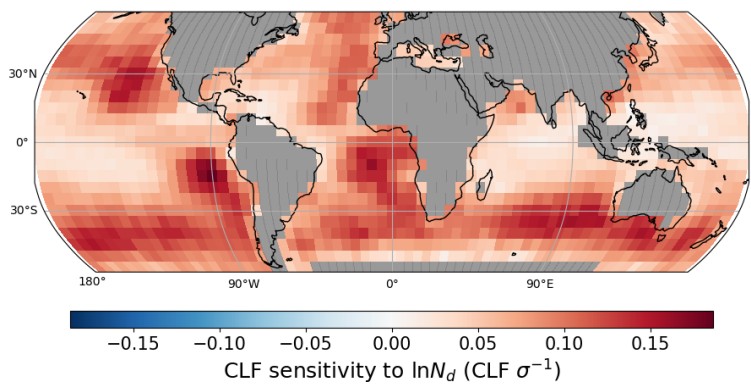

**Figure 4.** Sensitivity of marine boundary layer cloud fraction to $\ln N_d$.

the CLF-$\ln N_d$ sensitivity without standardization is 0.112 (unitless), and its spatial pattern is shown in Fig. S4. This value is higher than the upper bound of 0.1 reported by Bellouin et al. (2020), which is based on global climate models and large eddy simulations. This may be partly due to the aforementioned bias. However, it is important to note that our non-standardized CLF-$N_d$ sensitivity, shown in Fig. 1 (a), closely mirrors that from Yuan et al. (2023) with a similar range. In addition, the

high lnCLF–$\ln N_d$ values estimated in Chen et al. (2022b, 2024) suggest that values exceeding the upper bound of 0.1 might be plausible. These recent observational studies, including quantifying cloud fraction adjustment based on ship tracks Yuan et al. (2023), volcano aerosol perturbations (Chen et al., 2022b, 2024), and our SHAP approach using global satellite observations, indicate that the 0.1 upper bound may be extended. In future work, estimating a radiative forcing using the SHAP-based sensitivities will make our study more comparable with other research on cloud fraction adjustment.

### 3.2.3 Spatial patterns of the CLF sensitivity to thermodynamical drivers

There has been a strong consensus that EIS and SST are the two important determinants of cloud fraction of marine boundary clouds and their corresponding radiative effects across different geographical regions and on varying time scales (e.g. Bretherton, 2015; Myers and Norris, 2015; McCoy et al., 2017b; Wall et al., 2017). Stronger inversions capping MBL (i.e. higher EIS) will hamper the entrainment of aloft dry air from the troposphere and thus lead to a shallower MBL and more moisture trapped

within MBL, promoting the development and maintenance of low-level clouds (Andersen et al., 2017). The regional EIS–CLF sensitivity patterns (Fig. 5(a)) show that marine low cloud fraction increases ubiquitously in response to stronger EIS, in particular in the tropical and subtropical stratocumulus-capped regions and within the midlatitudes. The sensitivity pattern is in good agreement with that found by Scott et al. (2020) and Andersen et al. (2023), related studies at different time scales (Grise and Medeiros, 2016; Kelleher and Grise, 2019; de Szoeke et al., 2016).

MBLC cover reduces globally in response to increased SST, particularly pronounced in the stratocumulus regions over eastern oceanic basins (Fig. 5(b)), consistent well with (Scott et al., 2020). SST can favour MBLC dissipation through increasing surface latent heat fluxes and deepening MBL, facilitating dry entrainment and eventually desiccating the MBL and clouds

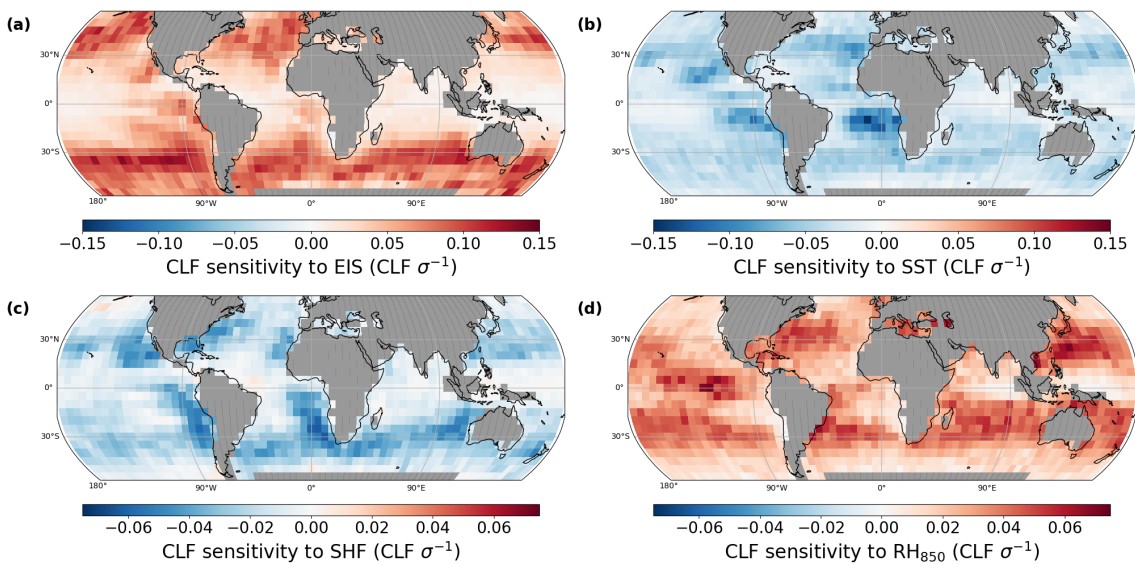

**Figure 5.** Sensitivity of marine boundary layer cloud fraction to the estimated inversion strength (EIS), sea surface temperature (SST), sensible heat flux (SHF) and relative humidity at 850 hPa (RH$_{850}$). Note that the range of colourbars of SHF and RH$_{850}$ (-0.075–0.075) is narrower than EIS and SST (-0.15–0.15).

(Rieck et al., 2012; Qu et al., 2015b). Yet as stated in Sect. 3.2.1, the weak CLF sensitivity to LHF in relation to the strong sensitivity to SST may imply that the other process makes more substantial contributions, namely that the higher moisture gra-

dient between the troposphere and MBL arising from the increased SST makes the entrained air more efficient in evaporating cloud water (van der Dussen et al., 2015; Qu et al., 2015b). This process has been shown to be the driving mechanism for the observed reduction in marine low cloud cover near the coast of Baja California (Andersen et al., 2022).

Figure. 5 (c) shows that low marine cloud fraction increases with negative (upward) SHF most markedly in the stratocumulus regions. CLF can increase in response to increased surface fluxes in situations of cold advection (Zelinka et al., 2018). Over

the South Indian Ocean, a marked SHF–CLF sensitivity is also found. Here, enhancements of SHF due to the subtropical anticyclone and midlatitude storm-track activity have been found to increase CLF (Miyamoto et al., 2018). The results may be a hint that the increase of CLF presumably due to increased SHF (e.g. due to cold advection) outweighs the influence of SHF on CLF by controlling the transition from marine stratocumulus to open-cellular marine clouds (Kazil et al., 2014; Fan et al., 2016) in the core stratocumulus regions. Consequently, the SHF–CLF sensitivity is less pronounced in regions of

frequent closed- to open-cell and cumulus transitions. Relative humidity at 850 hPa (RH$_{850}$) is positively related to marine low liquid cloud fraction across the globe. The positive sensitivity is particularly strong in the trade cumulus regions where the 850 hPa level is representative of the boundary layer. In the coastal stratocumulus regions, clouds are frequently below this level (Adebiyi and Zuidema, 2016), so that clouds are not as sensitive to variability in RH at that level.

### 3.2.4 Spatial patterns of the CLF sensitivity to dynamical drivers

Large-scale circulations and dynamical conditions play an essential role in controlling cloud fraction and the indirect effects of aerosols (Su et al., 2010; Small et al., 2011). The large-scale dynamics are represented by the horizontal and vertical winds at 700 hPa and 850 hPa, which display clear and distinct regional patterns of CLF sensitivity (Fig. 6). It can also be seen that at the considered scales and pressure levels, horizontal wind vectors have stronger CLF sensitivities than large-scale vertical motion. There is a coherent pattern of negative CLF sensitivity to the zonal wind at 700 hPa in the stratocumulus-dominated regions

(also apparent at 850 hPa), and the southern hemispheric midlatitudes, indicating a decrease in MBLCs with westerly anomalies at this pressure level. Recently, a study using monthly data has also found a similar sensitivity pattern of stratocumulus clouds to zonal wind at 700 hPa, finding that the reduced CLF is related to increased vertical wind shear (as the boundary layer flow is easterly), leading to increased turbulence and dry-air entrainment (Andersen et al., 2023). Using monthly data, Andersen et al. (2023) did not find a similar CLF sensitivity to zonal winds in the southern hemispheric midlatitudes, though. As the CLF

sensitivity to $u_{700}$ in the southern hemispheric midlatitudes is only apparent using daily data and only at 700 hPa, it seems likely that it is related to synoptic variability that drives day-to-day variability in MBLCs in this region (Kelleher and Grise, 2019). Positive CLF sensitivities to $u_{700}$ (higher CLF with westerly anomalies) and to a lesser degree $u_{850}$ are found off the eastern Asian and North-American continents. CLF increases due to cold-air outbreaks in NW Atlantic and NW Pacific may be the reason for these positive sensitivities. Cold-air outbreaks occur during winter as cold continental air moves over warmer

SSTs, increasing moisture and heat fluxes into the MBL so that the formation of MBLCs is favoured (Young et al., 2002). This leads to wintertime maxima in CLF in these regions (Yuan and Oreopoulos, 2013).

The sensitivity of CLF to the meridional winds at 700 hPa exhibits two bands straddling the subtropical regions between about 15 °and 35 °in both hemispheres but opposite in sign (positive in the Northern Hemisphere and negative in the Southern Hemisphere), illustrating that in these regions the poleward winds are associated with an increase in low cloud fraction. The

bands are still apparent at 850 hPa, while the negative band in the Southern Hemisphere extends northward to tropical areas. These hemispheric sensitivity bands to the v wind component at 700 hPa closely resemble those found in Andersen et al. (2023), with their analysis suggesting that the poleward winds on the eastern side of midlatitude cyclones may be related to warm and moist advection, increasing CLF. However, they also find a strong correlation of these free-tropospheric poleward winds with large-scale ascending air motion making the assertion of causality difficult. Poleward winds are also found to

decrease CLF over the southern hemispheric midlatitudes.

CLF is negatively connected to the vertical pressure velocity both at 700 hPa and 850 hPa ($\omega_{700}$ and $\omega_{850}$) over the entire Earth, indicating that ascending large-scale air motion enhances the cover of MBLCs globally. It is shown in the bottom of Fig. 6 column (a) that the CLF sensitivity to $\omega_{700}$ is larger in the midlatitude ocean basins, whereas the CLF sensitivity to $\omega_{850}$ is larger in the subtropical oceans where subsidence is climatologically prevalent (Myers and Norris, 2015, 2016; Scott et al.,

2020). This seems indicative of CLF being most sensitive to large-scale ascending motion at the typical altitude of the clouds. It is interesting to note that between 30°N and 30°S, no marked CLF sensitivity to $\omega_{700}$ is found, contrasting the finding of

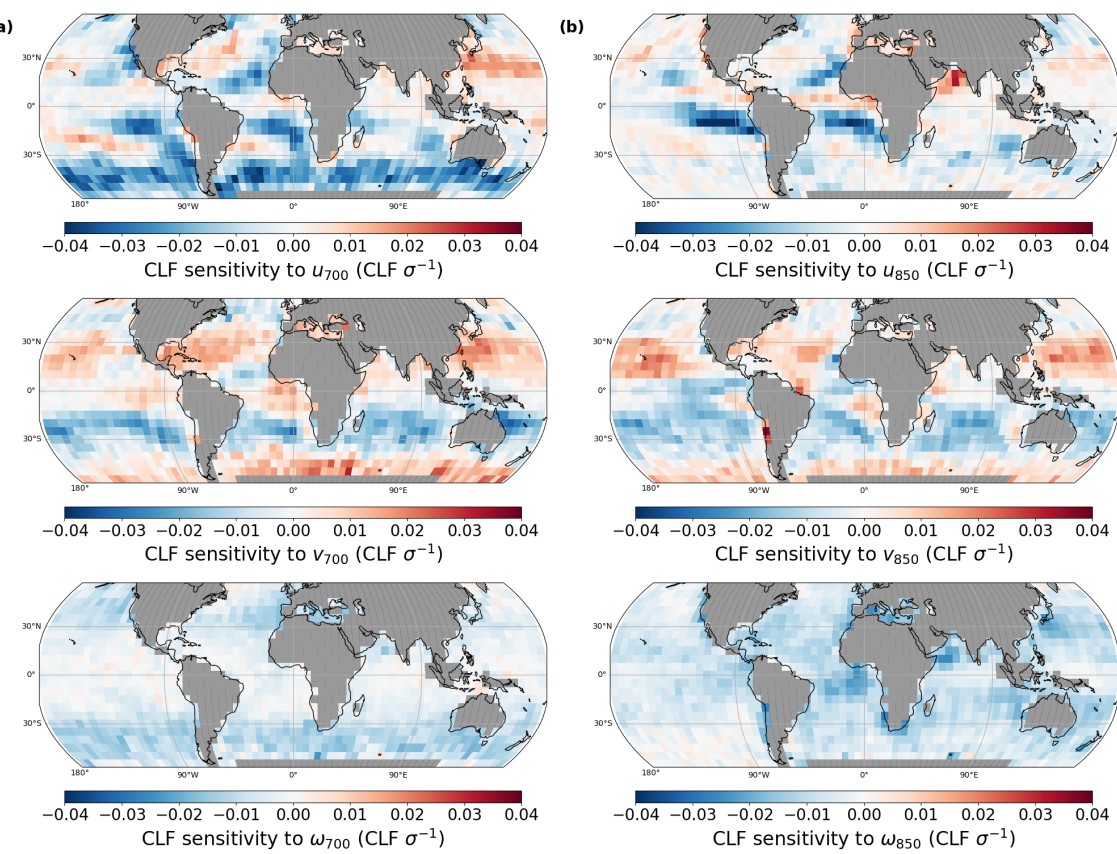

**Figure 6.** Sensitivity of cloud fraction to u, v wind component vectors and vertical velocities at 700 hPa (column (a)) and 850 hPa (column (b)). Note that the range of the colourbars is in general smaller (-0.04–0.04) than in Fig. 5.

enhanced subsidence at this level reducing MBLCs by Myers and Norris (2013). This effect is likely better described in the $\omega_{850}$ data which is more related to the altitude of the cloud top.

### 3.3 Dependence of $N_d$–CLF relationship on meteorology

### 3.3.1 Global overview of the Interaction Indices

In this section, we use the IAI as defined in Sect. 2.3.2 to quantitatively show how the response of MBLC fraction attributed to the aerosol proxy $N_d$ varies with the meteorological factors. As discussed in Sect. 2.3.2, since the sensitivity related to $N_d$ is positive across the globe (Fig. 5 (d)), a positive IAI can be interpreted as an amplification of the $N_d$–CLF sensitivity with high (above-average) feature values of a meteorological variable, whereas a negative IAI signifies an amplification of the sensitivity at low feature values.

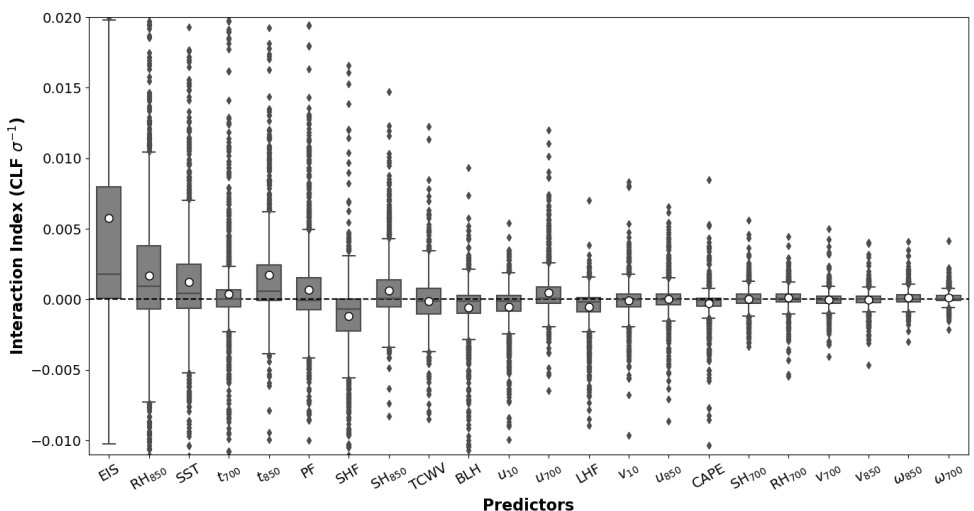

**Figure 7.** Similar to Fig. 3 but for the interaction effect of $N_d$ with all environmental parameters, quantified by the Interaction Index (CLF $\sigma^{-1}$).

In Fig. 7, analogous to Fig. 3, the features along the x-axis are arranged in descending order based on their averaged absolute IAIs i.e. by the strength of the impact of each meteorological feature on the $N_d$–CLF sensitivity. Similar to the feature importance summarized by Fig. 3, EIS, SST, $RH_{850}$ and SHF have relatively large strength of interaction effect and thus can be regarded as critical controlling factors not only for marine low cloud cover but also for their response to changes in $N_d$ (and
in extension aerosols). Compared to the CLF sensitivities, the IAIs associated with atmospheric temperatures at 700 and 850 hPa have greater strengths. Furthermore, it can also be seen that the vertical and horizontal winds at the surface and different pressure levels are ranked generally lower. In general, the thermodynamical factors seem to have a stronger influence on the $N_d$–CLF sensitivity than the dynamical factors.

### 3.3.2  Spatial patterns of the Interaction Indices

Coherent and distinct spatial distributions of the impact of selected meteorological parameters on the $N_d$–CLF relationship can be observed. Hereafter we show the regional characteristics of the interaction effects of EIS and SST which are the two most important meteorological factors for CLF in MBLCs and have the greatest absolute strengths of IAI. EIS exerts the most noticeable positive IAIs over the midlatitude oceanic areas (Figure. 8 (a)), reflecting that stronger temperature inversions capping the MBL over these regions may amplify the positive $N_d$–CLF relationship. The interpretation of possible underlying
physical mechanisms of these interaction effects is difficult and remains speculative. The results seem to suggest that in these regions, potentially through hampering the entrainment of drier air from the free troposphere, the stronger inversion and more stable conditions are capable of trapping more moisture within a shallower MBL and could thus weaken the evaporation-

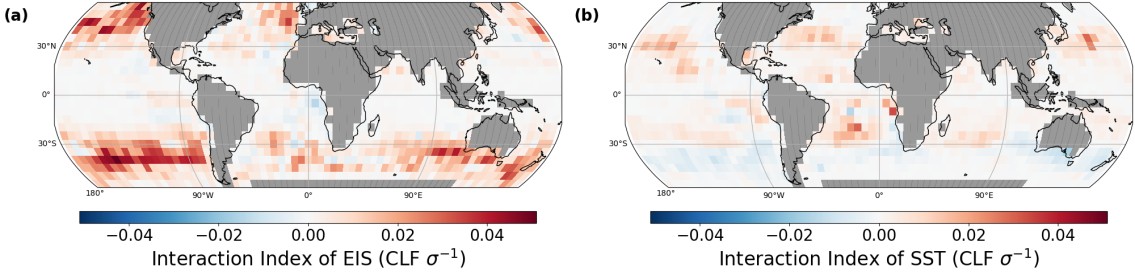

**Figure 8.** Patterns of the Interaction Index showing the dependence of the $N_d$–CLF relationship on estimated inversion strength (EIS) (a) and sea surface temperature (SST) (b).

entrainment feedback. As a result, it may ultimately favour a more positive $N_d$–CLF relationship (Chen et al., 2014; Christensen et al., 2020). It is interesting to note that these interactions are not apparent in the stratocumulus regions where EIS is a strong control of CLF, and in the stratocumulus-to-cumulus transition regions, where Christensen et al. (2020) found the aerosol effect on this transition to be confined to stable atmospheric conditions. This may imply that the suggested entrainment effect is dependent on the EIS, and stronger at slightly lower EIS values typically found in the midlatitudes (Scott et al., 2020). The observed impact of EIS on the $N_d$–CLF relationship found in the midlatitudes may also have implications within the context of climate change. While in the subtropics, global climate models predict an increase in EIS with a warming climate, in the midlatitudes EIS is predicted to decrease (Myers et al., 2021), potentially decreasing the sensitivity of CLF to $N_d$ there.

Fig. 8 (b) shows that higher SSTs are found to amplify the positive $N_d$–CLF relationship (positive IAI) in the regions of frequent stratocumulus-to-cumulus transition (Cesana and Del Genio, 2021). The physical interpretation could be: Here, higher SSTs tend to lead to the transition from stratocumulus clouds to shallow convective clouds (Cesana et al., 2019), however, this transition has been found to be delayed when aerosol is increased (Goren et al., 2019; Christensen et al., 2020). Tentatively, the positive IAIs in these transition regions may thus point to increased control of $N_d$ on CLF at higher SST values, as these are the situations where transitions typically occur and when increased $N_d$ can act to delay this transition. In these regions, higher SSTs in the future might thus increase the sensitivity of MBLC CLF to aerosols. It should be noted that the quantification of the dependence of the $N_d$–CLF relationship on meteorological factors (EIS, SST discussed in this section) is also likely subject to the biases in the $N_d$–CLF sensitivity caused by the $N_d$ retrieval biases as a function of CLF. This would potentially contribute to the non-causal facets of the relationships and interactive effects quantified by SHAP values.

## 4  Conclusions

In this study, nine years (2011-2019) of daily satellite and reanalysis data have been analyzed to better understand the effect of $N_d$ on CLF in MBLC, and its dependence on meteorological factors. We have established a near-global machine learning framework to predict the cloud fraction of marine boundary clouds using regionally-specific XGB regression models. Including many confounding and influencing factors as a whole, the explainable machine learning technique of SHAP regression values

has been used to explain the regional XGB models, to quantify the CLF sensitivity to all cloud controlling factors with a specific focus on $N_d$, moreover, to quantify the meteorological influence on the $N_d$–CLF relationship at a global scale. The statistical sensitivities and interactive effects are interpreted with the guidance of hypothesised causal pathways and the state-of-the-art physical understanding of the system. The main findings of this study, which should be interpreted in light of the data and methodology limitations discussed in Sect. 2.3.3), are summarized as follows:

1. Marine boundary layer cloud fraction shows a notable positive sensitivity to $N_d$ (a surrogate for aerosols) in the regions of stratocumulus to cumulus transition, which may arise from the high $N_d$ delaying this transition. The $N_d$–CLF sensitivity in the southern hemispheric midlatitudes is observed to be higher than in previous studies, which should be investigated in future work. The estimated $N_d$–CLF sensitivity and its magnitude suggest that aerosols likely have a considerable impact on MBL cloudiness, although this may partially result from an overestimation caused by the effect of a positive retrieval bias of $N_d$ at high CLF.

2. Consistent with the literature, our statistical method shows that EIS and SST are two important determinants for marine low clouds by regulating surface fluxes and dry-air entrainment processes. In addition, strong negative CLF sensitivity and spatial patterns for SHF are also found, suggesting that the effect of cold air advection might surpass the SHF enhancement of closed- to open-cell and cumulus transitions. Dynamic drivers (meridional and zonal winds) indicate that midlatitude synoptic-scale disturbances and vertical wind shear seemingly make considerable contributions to marine low cloud amounts.

3. In general, thermodynamical parameters exert a more important influence on the $N_d$–CLF relationship than dynamical parameters. EIS, $RH_{850}$, SST, temperatures at 700 and 850 hPa have the strongest effect on the $N_d$–CLF sensitivity. In the midlatitudes, higher EIS is found to amplify the positive $N_d$–CLF sensitivity which may be related to a reduced entrainment feedback in these conditions. Whereas higher SST is found to amplify the $N_d$–CLF sensitivity in stratocumulus-to-cumulus transition regions potentially because the transition induced by higher SSTs may be delayed by increased $N_d$. These findings have potential implications for possible future changes in the sensitivity of CLF to aerosols.

4. For the dynamical and thermodynamical factors shown here, both CLF sensitivities and the interactive effects (dependence of $N_d$–CLF relationship on meteorology) exhibit distinct regional patterns. These coherent spatial patterns indicate that the proposed explainable machine learning framework is not only capable of skillfully predicting CLF for marine low clouds but also has the potential to capture regional characteristics of the relation between CLF and $N_d$, as well as meteorological influences.

In the future, the observation-based sensitivities and interactive effects quantified by the ML framework here will be compared to those in ESMs, which have the potential to evaluate ESM parameterizations related to ACI and even help gain insights into how the models could be tuned in this respect. In addition, incorporating causal approaches for SHAP, such as proposed

by Heskes et al. (2020); Frye et al. (2021), would help to test the extent to which the observed statistical relationships and interaction effects represent physical processes.

*Code availability.* Code is available from the corresponding author on reasonable request.

*Data availability.* All data sets used in this study are publicly available. The MODIS data set (https://dx.doi.org/10.5067/MODIS/MOD08_M3.061) was acquired from the Level-1 and Atmosphere Archive & Distribution System (LAADS) Distributed Active Archive Center (DAAC) (https://ladsweb.modaps.eosdis.nasa.gov/, last access: 09 June 2021); The hourly reanalysis data at single levels (https://doi.org/10.24381/cds.adbb2d47) and pressure levels (https://doi.org/10.24381/cds.bd0915c6, last access: 22 Feb 2023) are obtained from the Copernicus Climate Change Ser-
vice (C3S) Climate Date Store: https://cds.climate.copernicus.eu/#!/search?text=ERA5&type=dataset.

*Author contributions.* HA and JC designed the initial research idea. YJ, HA and JC developed the study concept and methodology. YJ and HA obtained and analysed the data sets. YJ implemented the explainable machine learning framework, performed visualization and wrote the original draft. All authors contributed to interpreting the results, reviewing and improving the manuscript.

*Competing interests.* The authors declare that they have no conflict of interest

*Acknowledgements.* The (co-) authors have received funding from the European Union's Horizon 2020 research and innovation programme under grant agreement no. 821205 (FORCeS) and the Deutsche Forschungsgemeinschaft (DFG) in the project Constraining Aerosol-Low cloud InteractionS with multi-target MAchine learning (CALISMA), project number 440521482. We thank two anonymous reviewers whose helpful comments contributed to improving the manuscript.

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
