# Peer review of "Analysis of the cloud fraction adjustment to aerosols and its dependence on meteorological controls using explainable machine learning"

_EGUsphere, 2023_

## Author Comment (AC1)

**Response to the reviewers: Analysis of cloud fraction adjustment to aerosols and its dependence on meteorological controls using explainable machine learning # EGUSPHERE-2023-1667**

Yichen Jia [1,2], Hendrik Andersen [1,2], and Jan Cermak [1,2]

[1]Karlsruhe Institute of Technology (KIT), Institute of Meteorology and Climate Research, Karlsruhe, Germany
[2]Karlsruhe Institute of Technology (KIT), Institute of Photogrammetry and Remote Sensing, Karlsruhe, Germany

**Correspondence:** Yichen Jia (yichen.jia@kit.edu)

We would like to thank the two anonymous referees for reviewing the manuscript and providing very helpful and constructive comments. The reviewers' comments and suggestions are incorporated in italics and addressed hereafter and the authors' responses are colored in blue. If not explicitly stated otherwise, line numbers in this letter refer to the original manuscript (before the update). We have also added: "We thank two anonymous reviewers whose helpful comments contributed to improving the manuscript." to the acknowledgements.

**Referee 1**

**General comments**

*This is an interesting work focused on factors influencing the cloud fraction by using explainable machine learning approaches. The data and method are both solid, and the paper is well-written. I just found several places that need to be justified or clarified. Therefore, I recommend a minor revision for this paper to be published on ACP.*

Thank you for the positive evaluation of the manuscript. Specific comments are addressed individually as follows.

**Specific comments**

*Line 90: Does re means CLF? How cloud temperature, solar zenith viewing angle, and satellite zenith angle are used to filter and compute Nd?*

$r_e$ refers to cloud droplet effective radius (Section 1 Line 41). How cloud temperature, solar zenith viewing angle and satellite zenith angle are used to filter $N_d$ is described from line 105 to Line 109. To clarify, we rephrased the sentence "Except for CLF ... and compute $N_d$" in Line 90 as "CLF serves as the predictand in this study. The computation of $N_d$ relies on $\tau_c$ and $r_e$, with filtering criteria based on CTT, solar zenith viewing angle and satellite zenith angle, as elaborated in the following."

*Line 110: How the reanalysis data are harmonized? Could you please provide a little more details, which spatial averaging (interpolation) techniques have been used?*

We have revised the related sentence at Line 111-112 with a more detailed description: "The ERA5 data sets are harmonized to fit the level-3 MODIS data by first being resampled to $1° \times 1°$ from their default $0.25° \times 0.25°$ spatial resolution using bilinear

interpolation, and they are subsequently collocated to Terra MODIS by extracting hourly data to align with the UTC overpass times of the Terra satellite for each grid cell, yielding a spatiotemporally matched MODIS-ERA5 combined data set for training the ML models."

*Line 114: Why use 99 percentiles as the threshold? How extreme values will influence your interpretation of the ML model?*
We refined the sentences in Line 114 to explain this: "For $N_d$ retrievals, only samples within 1–99 percentiles are retained to exclude potential unrealistic outliers from $r_e$ and $\tau_c$ retrievals (Zipfel et al., 2022). Furthermore, the explanation of ML models in this study relies on using linear regressions to capture the distribution of individual prediction instances, and the extreme values may excessively magnify or reduce the sensitivity or interactive effects quantified by SHAP (shown in Fig. 1 and discussed in Sect. 2.3.2). The threshold of 1–99 percentiles for each predictor is thus adopted to remove the values at the very tails of the specific distribution and to improve the robustness of the estimated sensitivities."

*Line 136: 6000 data points as the threshold, would you please provide an estimate for the percentage of data dropped from the total sample?*
Thanks for the comment. The threshold of 6000 is used to guarantee that each $5° \times 5°$ "geographical window" has a sufficient volume of data to train and test the XGB models. As this study explicitly focuses on MBLCs, the $5° \times 5°$ windows encompassing continental or coastal regions and containing fewer than 6000 valid samples are not considered as "filtered out". For windows covering only oceans, 34 of them have fewer than valid 6000 samples due to the screening for $N_d$ retrievals. These specific windows, located between 47.5°W–122.5°E and 52.5°S–57.5°S, have been consequently removed.

We have modified the corresponding statement to further clarify this: "To ensure a sufficient data amount for training and testing the XGB models, only the geographical windows with over 6000 available data points are retained. Consequently, 34 out of 1190 oceanic windows have been excluded. These windows located between 47.5°W–122.5°E and 52.5°S–57.5°S in the Southern Ocean (Fig. 2) contain fewer than 6000 valid samples due to the screening for $N_d$ retrievals."

*Line 180: The definition of IAI may need to be clarified, is it a slope or difference? how the difference is calculated, which minus which?*
Thanks for the suggestion, the sentence has been revised for improved clarity as follows: "An interaction index (IAI) is derived from these regression fits and defined as the slope for the high-value group (> mean) minus the slope for the low-value group (< mean):

$$IAI = \beta_{x,high} - \beta_{x,low} \tag{2}$$

where $\beta$ is the slope of the linear regression between SHAP interaction values and ln $N_d$ values, and the subscripts denote the high-value group and the low-value group for a specific meteorological variable $x$ (SST in the example), respectively. At the exemplary geographical window, the influence of SST on $N_d$–CLF sensitivity is quantified by IAI = -0.029 CLF $\sigma^{-1}$ (Fig. 1 (b)). Similar to sensitivities, the unit of IAIs is also CLF $\sigma^{-1}$."

*Line 214: How do you explain that the PF is influencing the CLF but not vice versa? Same for some other predictors in the ML model.*

For PF, we intended to use the ML framework in this study to represent the role of large-scale precipitation on MBLC ACIs. As theoretical knowledge suggests atmospheric aerosols tend to affect cloud properties differently for precipitating clouds (precipitation suppression) and non-precipitating clouds (enhanced entrainment feedbacks) as outlined in the introduction section. It was assumed that the setup of IAI would be able to capture these effects.

Yes, CLF could also exert an influence on PF and other predictor parameters, and there may be relevant feedback loops at play. Similar to PF, the selection of all predictors, which could be potentially recognized as main drivers within the context of ACI, is based on the literature and existing knowledge for better interpretability of causality. The sensitivities and meteorological influences on ACI quantified by SHAP regression values are statistically representative of physical processes at regional and daily scales. Nevertheless, it is important to note that the causal chain quantified by SHAP values in the ML models may not necessarily reflect real-world physical causality, though this is an inherent and common limitation for all statistical analyses. We have added relevant discussions to the manuscript and rephrased it accordingly:

- P. 7 L. 165: "SHAP values provide insights into the behaviour of the XGB models, and as all statistical/ML models, they may not necessarily reflect real-world physical causality. Nevertheless, this state-of-the-art technique allows us to account for meteorological covariations when deriving sensitivities and to appraise to what extent the meteorological predictors interact with and influence the $N_d$–CLF relationship beyond traditional global-level feature attributions."

- P. 16 L. 352: "It should be noted ... $N_d$ retrieval biases as a function of CLF. This would potentially contribute to the non-causal facets of the relationships and interactive effects quantified by SHAP values."

*Line 229, SHF, please declare the full name of the acronym when it first appears in the text, even if it has been listed in table 1.*

Thanks for pointing it out, done. Full spellings of acronyms of other predictors have also been added when they are initially introduced in the main text. The use of full names and acronyms of all predictors has also been adjusted accordingly throughout the manuscript.

*Section 3.3.2, Fig. 8, why RH850 is omitted? It shows higher importance than SST in Fig.7.*

We want to emphasize the significance of EIS and SST, as they are critical components of the low-cloud feedback. RH can be regarded as an intermediate predictor as it can act as a proxy for various physical processes. The regional pattern of IAI for $RH_{850}$ is shown in Fig. 1 of this letter, in spite of the distinct patterns, the assertion of direct causality is challenging as relative humidity is linked to many meteorological processes.

[Figure]

**Figure 1.** Patterns of the Interaction Index showing the dependence of the $N_d$–CLF relationship on relative humidity at 850 hPa (RH$_{850}$).

**Referee 2**

**General comments**

*This manuscript fits xgboost models to capture daily cloud fraction at 5x5 degree scale. They apply the SHAP calculation and*
*present results mostly in terms of the SHAP values and their variations to different values by holding individual variables out*
*from model calculations. The authors argue that the results represent a 'better quantification of the responses of MBLC CLF to*
*aerosols'. The topic is relevant for ACP. However, there are a few major methodological concerns I have and they are detailed*
*in the following. In my opinion, they must be addressed before the paper can be published.*
We thank the referee for the comments which are very helpful in making the manuscript clearer. The concerns are addressed
individually as follows:

**Specific comments**

*1. It is unclear what data constitute MBL CLF. Only daily MODIS level-3 data are mentioned, but AFAIK, MODIS daily data*
*do not have a field called MBL CLF. The authors need to be clear on this.*
We have made the relevant statement clearer on P. 3 L. 88–90: "In this study, MBLCs are defined as single-layer warm cloud
fields with cloud top temperatures higher than 268 K. To achieve this, the information on CLF (product: Cloud_Retrieval_Fraction
_1L_Liquid), $r_e$ (product: Cloud_Effective_Radius_1L_Liquid_Mean), Cloud optical depth ($\tau_c$; product: Cloud_Optical_Thickness
_1L_Liquid_Mean), cloud top temperature (CTT; product: Cloud_Top_Temperature_Mean) and satellite viewing geometry are
obtained from MODIS level-3 collection-6.1 atmosphere daily products on the Terra platform (MOD08_D3), which are gridded
into $1° \times 1°$ globally from level-2 atmospheric products."

100    *2. It is unclear how the data is standardized. The standardization procedure is easy to understand, but is this done globally or locally for each 5x5 box?*

Standardization is done for each $5° \times 5°$ box before the training of each XGB model. We have added the information to the sentence on P. 4 L. 118 to clarify this: "All input data for each XGB model (i.e. for each $5° \times 5°$ window) are standardized for comparability of the estimates of the sensitivity and the interactive effect with meteorology (Sect. 2.3.2)."

105    *3. Some inappropriate reference citing is noted. For example, line 37, 'Furthermore, MBLCs are especially susceptible to aerosol perturbations due to their physical properties (Wood et al., 2015).' This is a very vague and general statement with a field campaign overview paper as a reference. As a researcher working on this topic for a long time, I cannot follow what is being said or cited here. Line 45: these two references are relevant, but they are neither the earliest nor the best papers that explain the mechanism mentioned here. Line 50: Yuan et al., 2011 showed increase of cloud fraction with aerosols in the trade*

110    *cumulus region. Line 52: observational evidence of GCM overestimation of LWP adjustment was presented convincingly in Toll et al., 2019. Line 102: potential retrieval biases are extensively discussed in Zhang et al., 2011. Grosvenor et al 2018 has relevant info, but it is not specifically for this subject.*

We thank the referee for bringing our attention to the references mentioned.

– Line 37: We agree that the choice of citing Wood et al. (2015) may not be the most suitable in this context. The sentence
115    has been refined for better clarity and precision regarding "their physical properties". It has been altered as follows: "Furthermore, MBLCs are especially susceptible to aerosol perturbations due to their relatively low optical depths (Turner, 2007; Leahy et al., 2012) and their formation in environments typically characterized by lower anthropogenic aerosol loading than continental clouds (Platnick and Twomey, 1994)"

– Line 45: Thanks for pointing this out. We have updated the references in question and included sedimentation-entrainment
120    feedback which can also lead to a decrease in clouds from increased $N_d$: "This has been found in particular for non-precipitating clouds, stemming from enhanced entrainment mixing with ambient air over the clouds owing to shorter evaporation timescales (Wang et al., 2003; Jiang et al., 2006; Small et al., 2009) or reduced sedimentation (Ackerman et al., 2004; Bretherton et al., 2007) because of smaller droplet sizes."

– Line 50: Thanks for the suggestion. Yuan et al. (2011) has been added to the manuscript.

125    – Line 52: We have added the suggested reference to Line 54 and rephrased the existing sentence: "This is supported by observational evidence presented by Toll et al. (2019) who also reported an overestimation of LWP adjustment in climate models, and by Chen et al. (2022) who recently highlighted the role of CLF increases due to aerosols from a large volcano eruption as the main cause of the associated forcing."

– Line 103: Thanks for pointing us to the earlier reference that discusses the potential $r_e$ retrieval biases in heterogeneous
130    clouds. We now cite Zhang and Platnick (2011) and Zhang et al. (2012) in the updated version of the manuscript: "In such heterogeneous cloud fields, subpixel effects in the retrieval of $r_e$ can negatively bias the retrieved $N_d$ values (Zhang and Platnick, 2011; Zhang et al., 2012; Grosvenor et al., 2018)."

*4. SHAP values, as the authors corrected noted, are only ONE way of attempting to explain the boosted tree models. For each data point, there is a SHAP value for each explaining variable. By construct, they are 'situationally' dependent. They don't really provide any physical insights. All the algorithm is trying to do is gradient boosting its model to best fit the data. In fact, the first figure shows that the way the authors try to use SHAP values to 'explain' results is not physical. Figure 1a shows that SHAP value for Nd generally gets larger with increasing Nd. Physically, it says that when clouds are more polluted, cloud fraction tend to increase with Nd more stronger. This runs against our physical understanding. Fitting a slope for the SHAP values and claiming this shows sensitivity of CLF to Nd are not valid IMO. I'd love to hear the authors' rationale here.*

We agree with the reviewer that SHAP values are just one way of explaining tree-based machine learning (ML) models, in fact, many of these methods are related. SHAP values are basically the partial dependence of y on $X_i$ but additionally include interactive effects for more complex models (as used here). These interactive effects make them 'situationally' dependent, which we exploit in our study. While we acknowledge that SHAP values provide insights into the behaviour of XGB models and do not necessarily reflect physical relationships, this limitation is true for any observational study that uses statistical/ML methods to try and better understand physical processes. It should be noted though that SHAP values are a well-established and state-of-the-art method to get insights on processes from observational data in ML frameworks (e.g. Lundberg et al., 2020; Stirnberg et al., 2021; Zipfel et al., 2022; Li et al., 2022). Relevant discussions have already been added to the manuscript and the existing text has been rephrased accordingly:

- P. 7 L. 165: "SHAP values provide insights into the behaviour of the XGB models, and as all statistical/ML models, they may not necessarily reflect real-world physical causality. Nevertheless, this state-of-the-art technique allows us to account for meteorological covariations when deriving sensitivities and to appraise to what extent the meteorological predictors interact with and influence the $N_d$–CLF relationship beyond traditional global-level feature attributions."

- P. 16 L. 352: "It should be noted ... $N_d$ retrieval biases as a function of CLF. This would potentially contribute to the non-causal facets of the relationships and interactive effects quantified by SHAP values."

Of course, we also agree with the reviewer that the ML model used just tries to best fit the data, but again, this is essentially true for any predictive statistical or ML model. It is just different mechanisms of fitting to the data that make them different. As such, we don't feel this is a valid criticism of using this method in this context.

We believe there is a misunderstanding with respect to the physical interpretation of the SHAP values for deriving the CLF sensitivities and the example of Fig. 1 (a) of the manuscript. "Physically, it says that when clouds are more polluted, cloud fraction tend to increase with Nd more stronger." This interpretation is not correct. The figure does show that the *contribution* of ln $N_d$ to the prediction of cloud fraction scales with ln $N_d$, which is of course also the case in a linear model (Y = mX + b). In fact, in a linear model, SHAP values are just the difference between the base value (average value) and the predicted y (see Fig. 2 of this letter, left panel). Positive (negative) SHAP values indicate that the specific feature value increases (decreases) the prediction compared to this base value. In other words, the base value is the reference point against which the contributions of individual features are measured, and the measure is SHAP values. If we designate this "reference point" as the zero point on the y-axis (subtracting the base value), we will get the y-axis in the right panel. In the case of a linear regression, the slopes

[Figure]

**Figure 2.** Figures taken from the SHAP documentation page, section "An introduction to explainable AI with Shapley values" on: https://shap.readthedocs.io/en/latest/overviews.html, last accessed 22 October 2023.

of regression models fit to the original data and the SHAP values are equal, it is just that the intercept of the line changes by the base value (as if the y-axis were shifted by the base value, cf. Fig. 2). As such, it is clearly valid to fit a line to SHAP values and interpret the slope as the sensitivity of y to $X_i$, and this is also the reason why the derived sensitivity patterns are so similar compared to the other satellite-based studies cited in the manuscript.

Figure 1 (a) of the manuscript illustrates that - in the considered geographical window - as $\ln N_d$ increases, the SHAP values for $\ln N_d$ consistently increase (see the dashed line). Hence, the correct physical interpretation of Fig. 1 (a) within the manuscript is that as clouds become more polluted, the predicted CLF exhibits a nearly linear increase. which is in agreement with our physical understanding. The advantage of this method is twofold: 1) meteorological covariates are considered in the quantification of SHAP values and thus accounted for in the derived sensitivities, 2) possible effects of 3rd variables on the $N_d$–CLF relationship can be analyzed with interactive effects.

To make the manuscript clearer, we also have expanded and provided a more comprehensive explanation of SHAP values on P. 7, L. 155–157: "The base value in the context of SHAP values is typically computed as the average of all predictions by ML models over the entire training data points. Positive (negative) SHAP values indicate that the specific feature value increases (decreases) the prediction compared to this base value. In other words, the base value serves as the reference point against which the contributions of individual features are measured. SHAP values for all features will always sum up to the difference between the base value and the final model prediction so that SHAP values are additive and internally consistent."

*5. What follows in the manuscript is thus questionable. I will reserve my comments for the next version after the authors address the important methodological question.*

We believe that the questions and comments have been properly addressed.

**Minor modifications independent of the reviewer comments**

P. 3 L. 79: "SHAP" appears for the first time in the main text, the full spelling has been added here.

P. 4 L. 97: The blank between 50 and % has been deleted.

P. 4 L. 110: "20" has been removed from the text.

P. 6 L. 152: Addition reference (Li et al., 2022) has been added.

P. 7 L. 159: "when" is added between "versus" and "it".

P. 7 L. 172: "isolated" is now removed from the text.

P. 7 L. 173: the unit of the sensitivity (CLF $\sigma^{-1}$) has been added after 0.098. The sentence originally in L. 184 "Note that all the input data have been standardized ... standard deviation (CLF $\sigma^{-1}$)." has been restructured and split into L. 174 and L. 181, respectively, after mentioning the values of sensitivities and IAIs in the example.

P. 8 Figure 1: We added the unit of the sensitivity and IAI (CLF $\sigma^{-1}$) to the legends of subplots.

P. 11 L. 238: "Sec. 2.1" to "Sect. 2.1"

**References**

[revised manuscript text omitted]

---

## Author Response (AR2)

**Response to the reviewers: Analysis of the cloud fraction adjustment to aerosols and its dependence on meteorological controls using explainable machine learning   # EGUSPHERE-2023-1667**

Yichen Jia [1,2], Hendrik Andersen [1,2], and Jan Cermak [1,2]

[1]Karlsruhe Institute of Technology (KIT), Institute of Meteorology and Climate Research, Karlsruhe, Germany
[2]Karlsruhe Institute of Technology (KIT), Institute of Photogrammetry and Remote Sensing, Karlsruhe, Germany

**Correspondence:** Yichen Jia (yichen.jia@kit.edu)

We would like to thank the two anonymous referees for their reviews of the revised manuscript and their insightful feedback. Below, the reviewers' comments and suggestions are incorporated in italics and addressed hereafter, and the authors' responses are coloured in blue. Unless otherwise stated, line numbers in this document refer to the manuscript after the first-round review (before the updates following in this response letter).

**Referee 1**

**General comments**

*I appreciate the effort and work the authors have done to address my questions and issues raised in the last round. The manuscript has significantly improved and almost is ready for publication,*
Thank you for the kind words and the positive evaluation of the manuscript.

**Technical issues**

*1. line 326, two "?" I do not know what they mean.*

*2. line 339, same issue. Please double-check all references and format, etc.*
Thank you for catching that. This issue due to a wrong citation key has been corrected throughout the manuscript.

**Referee 2**

**General comments**

*On the point of justifying SHAP values of a complicated tree model, the authors make the point that basically everything statistical modeling technique is doing similar things. While that to some extent is true, it is besides the point. The point is that using extremely complicated models, that is beyond say individual scientists' grasp, to 'understand' behavior of low clouds is questionable. The understanding in this context comes from calculating SHAP values of a highly nonlinear model. While SHAP values are popular, they have their own limitations: the calculation itself depends on the software package, it assumes feature*

*independence, and most importantly these calculations are not meant for causal inference. That is precisely I made the point of checking the 'physical sense' of model behavior reported here. I used figure 1a as an example. As $N_d$ increases, the SHAP values continue to scale with $N_d$ (or lnNd). In other words, lnNd contributes to increase low cloud fraction linearly. This is not physical and not what we observe in nature. If anything, multiple lines of evidence suggest that $N_d$ tend to saturate after a*

25 *rather low threshold (Rosenfeld, et al., 2012; 2019; Yuan et al., 2023). This behavior is determined by low cloud precipitation formation and the threshold is quantified to be around 60-80 per CC. Similarly, I mentioned systematic biases associated with $N_d$ calculations and low cloud fraction. This is also physical and built in with the data the authors used. In other words, there is a strong component of covariance between these two that does not have physical implications but artificial correlation. These are first order considerations when attempting to 'explain' low cloud fractions. The standardization procedure is also puzzling*

30 *since it means that all the quantities calculated here are based on local data distribution and thus not really comparable between different 5x5 grids. It is however fine for the authors to try to analyze the data and report their statistical findings. It may be a useful endeavor for readers of this paper. However, it is clear to me that the authors do not seem to have a solid background in the physical understanding of low cloud formation and aerosol effects on them while attempting to extract physical explanations from statistical models. To this point, there is a saying by Einstein that says 'Everything should be made*

35 *as simple as possible, but not simpler'. The authors complied a long list of variables and used them to best fit the data. I suggest the authors to write the paper based on actual data and do not make too much into causal inference and discuss physical implications because the tool used here does not allow that. Particularly, the results do not always make physical sense anyway*

We thank the reviewer for her/his input, and see three main points that the reviewer is concerned about: 1) model complexity,

40 SHAP limitations, biases in the data, 2) potentially unphysical relationships, and 3) standardization procedure. In the following, we address these three concerns.

1. While in the original version of the manuscript, we already discussed the potential issues related to retrieval biases leading to positive $N_d$–CLF correlations in the data section, we take the concerns of the reviewer seriously, and have added a new section (Sect. 2.3.3, titled "Limitations of observation-based machine-learning of aerosol-cloud processes")

45   dedicated to discussing the limitations of this study regarding the data sets and methods used. Within this section, we discuss the general limitation of the SHAP approach, the built-in limitation of the data sets, the somewhat limited interpretability of the method with regard to physical processes, and the appropriate interpretation of sensitivities in this study. Many of these limitations are not limited to this study, though, but rather general limitations for observation-based machine-learning of aerosol-cloud processes (or more specifically the cloud fraction adjustment). In this section, we now

50   explicitly state that all these limitations should be considered when interpreting the results. To be more careful with the interpretation of the results, we have also removed "in a physically meaningful way" from the last sentence in the fourth point of the conclusion section. Another outlook point has been included at the end of the manuscript: "In addition, ... processes."

[Figure]

**Figure 1.** The left panel is the same as Figure 1 (a) in the manuscript. The orange line represents the rate of change in SHAP values for $N_d$ due to the $N_d$ change (dSHAP/d$N_d$) across binned $N_d$ values. The right panel is taken from Yuan et al. (2023). The solid blue line shows the rate of CLF change with respect to $N_d$ change (dCF/d$N_d$) as a function of binned $N_d$. Here we highlight the similarity between the orange line in the left panel and the blue line in the right panel.

2. Unphysical relationships: The reviewer states that "As $N_d$ increases, the SHAP values continue to scale with $N_d$ (or $\ln N_d$)", and later mentions that the effect of $N_d$ on CLF saturates at low $N_d$ values ($\sim$ 60–80 per CC), as e.g. found in Yuan et al. (2023). We agree to the latter statement. However, it is important to point to the difference between the CLF–$N_d$ and CLF–$\ln N_d$ relationships. In Yuan et al. (2023), the $N_d$–CLF relationship (dCF/d$N_d$) is shown to be sensitive at very low $N_d$ values and then approaches 0 in the value range mentioned by the reviewer. We see the same behaviour in our data sets (see Fig. 1 in this response), which is why we use the natural logarithm of $N_d$ to create a linear relationship. We have added the left panel of Fig. 1 of this response as an additional new panel to the first figure of the manuscript to show this difference between the SHAP dependence plots for $N_d$ and $\ln N_d$. As shown in the previous version of the manuscript, $\ln N_d$ SHAP values continue to scale linearly with $\ln N_d$, while the increase in $N_d$ SHAP values due to increasing $N_d$ saturates at higher $N_d$ values (for this specific geographical window, the sensitivity approaches 0 at about 100 per CC). This behaviour holds for not only this specific example geographical window, but also for others in general. Overall, the development of the CLF sensitivity to $N_d$ as a function of $N_d$ behaves remarkably similar when compared to Yuan et al. (2023) (see Fig. 1 in this response for a direct comparison). This implies that the use of SHAP values in explaining XGB captures the physical relationships similar to the study mentioned by the reviewer. We thus reject the notion of the reviewer that the central physical relationships discussed in our manuscript are unplausible, but rather that the comparison to the study mentioned by the reviewer supports our findings. We do see though that the communication of these results was not ideal but believe that the new panel in Fig. 1 in the manuscript helps the reader better understand this important aspect of our study. Section 2.3.2 has also been revised accordingly.

3. The z-score transformation of data sets is a typical procedure for studies also aiming to compare cloud responses to multiple different meteorological variables (e.g. Scott et al., 2020; Andersen et al., 2023). The advantage of this preprocessing step is that it places all meteorological variables on the same scale allowing a comparison to a typical variation in each meteorological field. We acknowledge that the downside of this approach, i.e. that standard deviations change regionally, making the interpretation of the spatial patterns more difficult, and that readers may be interested in the sensitivities from the non-standardized data. To address this, we have added a supplementary material showing the results from non-standardized data. We can clearly see that, in the supplement, the example SHAP dependence plots and the geographical patterns of sensitivities and IAIs are nearly identical to the corresponding ones in the manuscript, only absolute magnitudes are notably different as expected. The last part of Sect. 2.1 has been rewritten and expanded. Relevant discussions have also been added in Sect. 2.3.2.

**Minor modifications independent of the reviewer comments**

"the" is added before "cloud fraction" in the title.

Space between "ln" and "$N_d$" is removed throughout the text of the manuscript (ln $N_d$ to ln$N_d$).

P. 5 L. 130: "(see Sect. 2.3.2 in detail)" has been added after "logarithm of $N_d$ is taken".

Figure 5 caption: "are" to "is".

Figure 6 caption: "colourbar" to "colourbars".

P. 17 L. 386: "Including many confounding and influencing factors as a whole," has been added before "The explainable machine learning technique ...".

P. 17 L. 389: we have added a sentence "The statistical sensitivities ... physical understanding of the system." before "The main findings of ... as follows:".

**References**

Andersen, H., Cermak, J., Douglas, A., Myers, T. A., Nowack, P., Stier, P., Wall, C. J., and Wilson Kemsley, S.: Sensitivities of cloud radiative effects to large-scale meteorology and aerosols from global observations, Atmospheric Chemistry and Physics, 23, 10 775–10 794, https://doi.org/10.5194/acp-23-10775-2023, 2023.

Scott, R. C., Myers, T. A., Norris, J. R., Zelinka, M. D., Klein, S. A., Sun, M., and Doelling, D. R.: Observed Sensitivity of Low-Cloud Radiative Effects to Meteorological Perturbations over the Global Oceans, Journal of Climate, 33, 7717–7734, https://doi.org/https://doi.org/10.1175/JCLI-D-19-1028.1, 2020.

Yuan, T., Song, H., Wood, R., Oreopoulos, L., Platnick, S., Wang, C., Yu, H., Meyer, K., and Wilcox, E.: Observational evidence of strong forcing from aerosol effect on low cloud coverage, Science Advances, 9, eadh7716, https://doi.org/10.1126/sciadv.adh7716, 2023.

95

100

---

## Author Response (AR3)

**Response to the reviewers: Analysis of the cloud fraction adjustment to aerosols and its dependence on meteorological controls using explainable machine learning # EGUSPHERE-2023-1667**

Yichen Jia [1,2], Hendrik Andersen [1,2], and Jan Cermak [1,2]

[1]Karlsruhe Institute of Technology (KIT), Institute of Meteorology and Climate Research, Karlsruhe, Germany
[2]Karlsruhe Institute of Technology (KIT), Institute of Photogrammetry and Remote Sensing, Karlsruhe, Germany

**Correspondence:** Yichen Jia (yichen.jia@kit.edu)

We thank the anonymous referee for the new round of review of the revised manuscript. Please see the main points that the reviewer is concerned about. Below, the reviewer's comments and suggestions are incorporated in italics and addressed hereafter, and the authors' responses are coloured in blue. Unless otherwise stated, line numbers in this document refer to the manuscript after the second-round review (before the updates following in this response letter).

5 **Referee 2**

**Specific comments**

1. *Acknowledging the issues that are fundamental to the xgboost based approach, i.e., variable independence and potential artificial correlation is a good first step. But it does not address the important point.*

   Thank you for your feedback, we believe the manuscript has improved with the inclusion of a separate section discussing
10   the method and data limitations.

2. *It remains unphysical. First of all, SHAP value is already kind of a sensitivity of the target value to the dependent variable. That is, SHAP value of CF is equivalent to dCF/dNd. Second of all, the SHAP value for Nd and its dependence on Nd figure the authors showed in the response. even if we forget the first point, are qualitatively different from the figure from the reference. Their shape is similar, which is true. However, the SHAP value turns to strongly negative*
15   *values, which would be interpreted as CF decreases with Nd at these Nd values. That is unphysical either. I could name other physically inconsistencies if the authors show more details like this. The overaching point remains that we do not have reason to believe such boosted tree models would necessarily give us physical insights. I'd have not issues with authors publishing it as a statistical analysis, but if physical interpretations are involved the authors need to demonstrate them with care first.*

20   We thank the reviewer for his/her comments. However, the assertions made by the reviewer concerning SHAP values are incorrect or inaccurate. The use of our method aligns with the design of SHAP values, a similar way of sensitivity estimation was also applied in a Nature Communications paper (Li et al., 2022). To clear up the confusion, our response therefore addresses each point made by the reviewer separately in the following:

(a) *It remains unphysical. First of all, SHAP value is already kind of a sensitivity of the target value to the dependent variable. That is, SHAP value of CF is equivalent to dCF/dNd.*

**Short answer:** The interpretation of SHAP values as sensitivity and them being comparable to dCF/dNd is incorrect. A better analogy would be that SHAP values are comparable to $dCF/dN_d \times N_{dj}$, where $j$ denotes the $N_d$ value for a specific data instance $x_j$. Other studies (e.g., Li et al., 2022) have employed the same sensitivity estimation strategy as we have done in our paper.

**Longer answer:** SHAP values quantify feature contributions for data instances (feature values), they are not a sensitivity estimate (see e.g. the paper from the developer (Lundberg et al., 2020) or this textbook on explainable machine learning (Molnar, 2022)). Regarding the specific mention of SHAP values for $N_d$ (we assume the reviewer was referring to $N_d$ because there are no "SHAP values of CF"), it should be noted that they are neither equivalent nor directly comparable to $dCF/dN_d$. SHAP values do not directly quantify the rate of change of the target value with respect to changes in the dependent variables. Instead, as we already explained in the manuscript (in lines 175–181), and our response to the first review by R2, SHAP values represent the contribution of each feature to the CF prediction (compared to a base value) for individual data instances (i.e. local contribution). In fact, SHAP values are more comparable to $dCF/dN_d \times N_{dj}$, where $j$ denotes the $N_d$ value for a specific data instance $x_j$.

Mathematically, SHAP values are derived from Equation (1), wherein, for a given non-zero subset $S$ of feature values, the prediction is assumed to be equal to the expected value of the function conditioned on $S$ ($f_x(S) = E[f(x)|x_S]$) (Lundberg and Lee, 2017; Lundberg et al., 2018).

$$\phi_i = \sum_{S \subseteq N \setminus \{i\}} \frac{|S|!(M - |S| - 1)!}{M!} [f_x(S \cup \{i\}) - f_x(S)] \tag{1}$$

Equation (1) combines the traditional equation for Shapley values with conditional expectations. $N$ is the set of all features, $i$ denotes the $i$-th feature, $M$ is the number of input features and $f$ is the ML model function.

Because SHAP values do not provide a sensitivity estimate, we quantify the sensitivity by fitting a linear regression to the feature values and their respective SHAP values. This method has been used before, e.g. by Li et al. (2022) in their Nature Communications paper. In this study, the authors estimate the sensitivity of leaf area index (LAI) to sub– or near–surface soil moisture (SMsurf) as the slope of the regression between feature values of SMsurf anomalies and their SHAP values (Fig. 1 of this letter). The authors further argue that this approach facilitates the robustness of the sensitivity estimation compared to traditional statistical methods because it "combines the advantages of bootstrap aggregating and non-distribution-assumption by random forest modeling, as well as advantages of global interpretations being consistent with the local explanations in the SHAP algorithm (Li et al., 2022, page 7, section on overall sensitivity)." Our study also benefits from similar advantages and enhanced robustness. We have referenced this study and included relevant discussion in Sect. 2.3.2 of the manuscript at line 212.

(b) *Second of all, the SHAP value for Nd and its dependence on Nd figure the authors showed in the response. even if we forget the first point, are qualitatively different from the figure from the reference. Their shape is similar, which*

[Figure]

**Figure 1.** Figure adapted from the supplementary information of (Li et al., 2022), showing the sensitivity of leaf area index (LAI) to sub– or near–surface soil moisture (SMsurf) as the slope of the Theil-Sen linear regression between SHAP values and feature values.

*is true. However, the SHAP value turns to strongly negative values, which would be interpreted as CF decreases with Nd at these Nd values.*

**Short answer**: Again, the interpretation of the SHAP values by R2 is not correct, and thus only the acknowledgement of the similarities between our results and the findings by Yuan et al. (2023) remains.

**Longer answer**: It is incorrect to interpret negative SHAP values for $N_d$ at low $N_d$ values as CF decreases with $N_d$ (i.e. negative sensitivity). As we already explained in Sect. 2.3.1 of the manuscript in lines 177–180, as well as in our first response to the first review of R2, SHAP values indicate that the specific feature value increases/decreases the prediction compared to the "base value" (average of all predictions). SHAP values therefore quantify the extent to which each feature contributes to a prediction deviating from the model's average prediction/baseline. Fig. 2 from Lundberg et al. (2018) shows an example plot of how the sum of the baseline value and all feature contributions (positive and negative SHAP values) is equal to the individual model prediction. These feature contributions depend on the feature value. In the example $E\left[f\left(x\right)\right]$ is the base value (the prediction if we did not know any information on the input features), and $f\left(x\right)$ is the current model output. This plot illustrates how positive SHAP values (red) push the model prediction higher and negative SHAP values (blue) push the model prediction lower. This also indicates an important internal consistency of SHAP: $\sum_{i=0}^{M}\phi_i = f\left(x\right)$, where $i = 0$ denotes what would be predicted in the absence of any feature information (base value).

Figure 3 (which we have already shown in our first response to R2) shows the simplest case of SHAP values when they are applied to a linear model. The slopes of linear regressions fitted to the original data and the SHAP values are equal. The only difference is between the y-axes of $E\left[f\left(x\right)\right]$, as the base value is subtracted in the case of the SHAP values. This reaffirms the validity of using the linear regression slope of feature values and SHAP values for sensitivity estimation, but also clearly shows how R2 misinterprets the negative SHAP values at low $N_d$ as a negative sensitivity. In the case of a positive linear sensitivity low feature values (in Fig. 3 MedInc) will lead to

[Figure]

**Figure 2.** Illustration of how SHAP values explain the output of a function f as a sum of the effects $\phi_i$ of each feature being introduced into a conditional expectation (Lundberg et al., 2018).

[Figure]

**Figure 3.** Example figures showing the application a linear regression (left) and its SHAP values (right) - the only difference in this case is the subtraction of the base value (horizontal dashed line of $E\left[f\left(x\right)\right]$ in the left panel). Figures are from the SHAP documentation page, section "An introduction to explainable AI with Shapley values" on: https://shap.readthedocs.io/en/latest/overviews.html, last accessed 03 March 2024.

a below-average prediction of y (left panel), which in the case of SHAP values is a negative value (right panel). Therefore, negative SHAP values "decrease" the prediction with respect to the base value (i.e. at very low $N_d$ values a below-average CF is expected). In our study, the correct interpretation of local/individual SHAP values is provided in lines 177-179: "Positive (negative) SHAP values indicate that the specific feature value increases (decreases) the prediction compared to this base value.", and for the example of the global interpretation of $N_d$ SHAP values in Fig. 1a) is given in line 201: "...increased $N_d$ values lead to an increase in the predicted CLF, while the rate of the increase ($dSHAP/dN_d$) drops with $N_d$ as shown by the orange line." This is physically expected and agrees well with the cited literature.

**In summary, SHAP values are not a sensitivity measure. Sensitivity should be interpreted from a global perspective, whereas SHAP values should be initially interpreted from a local viewpoint, and then aggregated and summarized by a global sensitivity (here the slopes of the linear regressions, as also done in e.g. Li et al. (2022)), or global feature importance (the mean of absolute local SHAP values e.g. Fig. 3 of the manuscript).**

**This is well summarized in the explainable machine learning textbook by Molnar (2022): "The global interpretation methods include feature importance, feature dependence, interactions, clustering and summary plots. With SHAP, global interpretations are consistent with the local explanations, since the Shapley values are the "atomic unit" of the global interpretations (section 9.6.10)."**

(c) *That is unphysical either. I could name other physically inconsistencies if the authors show more details like this. The overaching point remains that we do not have reason to believe such boosted tree models would necessarily give us physical insights. I'd have not issues with authors publishing it as a statistical analysis, but if physical interpretations are involved the authors need to demonstrate them with care first.*

The concerns raised regarding unphysical $N_d$–CF relationships or physically inconsistent ones are based on the reviewer's misinterpretation of SHAP values, but are shown to be physically consistent and in agreement with the literature. We certainly agree with the reviewer that the interpretation regarding physical processes should always be done with care, (no matter if using a linear regression, a neural network or boosted trees) which we have done by:

  i. In the abstract, now explicitly stating that the results are based on a statistical/data-driven method and mentioning that limitations are discussed in the main text.

 ii. Openly discussing the potentials and limitations of the method and the data (e.g. in Sect. 2.3.3).

 iii. Directly stating in line 273 that the interpretation should be done with the limitations in mind.

 iv. Careful wording when physical interpretations are made (e.g. "These marked positive $N_d$–CLF sensitivities *may be* caused by high $N_d$ delaying the transition from stratocumulus to cumulus clouds (Gryspeerdt et al., 2016; Christensen et al., 2020)", line 332).

  v. We now additionally rephrase the manuscript including: add "seems to" before "indicates" at line 310. Line 324: "leading to" to "which could lead to". Line 361 and 362: "suggests that" to "may be a hint that"; add "presumably" after "increase of CLF". Line 417: add "seem to" between "factors" and "have".

 vi. Directly following up such interpretations by detailing possible unphysical alternative explanations (e.g. "As $N_d$ retrievals tend to negatively bias at lower CLF and positively bias at higher CLF, the $N_d$–CLF sensitivity may be overestimated, and at the scales considered here, should be interpreted as an upper bound to the physical $N_d$–CLF sensitivity.", lines 337–339).

 vii. Emphasizing again that the sensitivities and IAIs are subject to aforementioned limitations: "should be noted that the quantification of the dependence of the Nd–CLF relationship on meteorological factors (EIS, SST discussed in this section) is also likely subject to the biases in the Nd–CLF sensitivity caused by the Nd retrieval biases as a function of CLF. This would potentially contribute to the non-causal facets of the relationships and interactive effects quantified by SHAP values.", lines 441–444.

viii. We have also rephrased Sect. 3.3.2 to underscore the caution in making physical interpretations.

125       ix. In the conclusion section, it has been mentioned that "The statistical sensitivities and interactive effects are interpreted with the guidance of hypothesised causal pathways and the state-of-the-art physical understanding of the system." (lines 451–453); we also mentioned incorporating causal setups for SHAP would be a promising way to go (lines 476–478). Furthermore, we revised the conclusion part to exercise caution and to serve as a reminder to readers.

130   3. *This proves my point. The transformation basically doesn't make sense. Nearly all figures and results in this paper are about gross statistics and map distributions. When the underlying statistics are not consistent, it has to be corrected IMHO.*

The use of standardized regression coefficients is a standard and common practice when aiming for comparability of sensitivity estimates among predictors. It is described and recommended as the standard procedure to eliminate the
135 effect of units and place the predictors on the same scale in this paper: "A review of techniques for parameter sensitivity analysis of environmental models" (Hamby, 1994). As it is a general strategy to compare sensitivities across predictors, there are plenty of environmental studies that employ this strategy. We realize that the reviewer probably takes an issue with the resulting maps (standard deviations are not the same everywhere), rather than with the technique itself. We have revised the content introducing the standardization in lines 132–138 and have included a more direct discussion about
140 the trade-off between comparability between predictors, vs. comparability in space into the manuscript. One should note though, that even in this context (maps of sensitivities) this is still a standard method. Examples can be found e.g. in this nature paper (Seddon et al., 2016), which quantifies and shows maps of standardized sensitivities for multiple predictors. In the cloud community it is also commonly done when sensitivities are compared among predictors, and in similar settings (papers showing standardized sensitivity maps). Here is an incomplete list of references that have opted for this
145 strategy:

    – (Scott et al., 2020): Journal of Climate paper that quantifies standardized low-cloud sensitivities for a range of cloud-controlling factors, please see Fig. 4 of this letter for examplary sensitivity maps (note the unit of the color bar).

    – (Myers et al., 2021): Nature Climate Change paper that uses the standardized low-cloud sensitivities from Scott
150        et al. (2020).

    – (Ceppi and Nowack, 2021): PNAS paper that quantifies standardized cloud sensitivities for a range of cloud-controlling factors, as shown by Fig. 5 of this letter for examplary sensitivity maps.

    – (Andersen et al., 2023): ACP paper that quantifies standardized sensitivities of cloud radiative effects to a range of cloud-controlling factors including aerosols.

155     – (Grise and Kelleher, 2021): Journal of Climate paper that quantifies standardized sensitivities of cloud radiative effects in the midlatitudes for a range of cloud-controlling factors, Fig. 6 of this letter for examplary sensitivity maps.

[Figure]

**Figure 4.** Figure taken from Scott et al. (2020) showing standardized low-cloud sensitivities to different cloud-controlling factors.

- (Wilson Kemsley et al., 2024): EGUsphere manuscript that quantifies standardized high-cloud sensitivities for a range of cloud-controlling factors.

We do not share the opinion of the reviewer that this method "does not make sense", and the published literature in the cloud community (but also more broadly in environmental sciences) supports this assessment. As we believe the added value of the comparability between the sensitivities of the different predictors is of interest to the readership of ACP, and outweighs the discussed downside of marginally reduced comparability in space (as shown by the supplementary material), we would like to keep the figures in the manuscript to show standardized sensitivities with the non-standardized sensitivities in the supplement (as done in Grise and Kelleher (2021)).

**Minor modifications independent of the reviewer comments**

Abstract, line 20: "ACIs" to "the CLF adjustment".
Line 20: "-" between "CLF" and "sensitivities" has been removed.

[Figure]

**Figure 5.** Figure from Ceppi and Nowack (2021) showing shortwave cloud-radiative sensitivities to standardized surface temperature and EIS.

[Figure]

**Figure 6.** Figure from Grise and Kelleher (2021) showing sensitivities of low and high cloud fraction to four different standardized cloud-controlling factors.

**References**

170    Andersen, H., Cermak, J., Douglas, A., Myers, T. A., Nowack, P., Stier, P., Wall, C. J., and Wilson Kemsley, S.: Sensitivities of cloud radiative effects to large-scale meteorology and aerosols from global observations, Atmospheric Chemistry and Physics, 23, 10 775–10 794, https://doi.org/10.5194/acp-23-10775-2023, 2023.

Ceppi, P. and Nowack, P.: Observational evidence that cloud feedback amplifies global warming, Proceedings of the National Academy of Sciences, 118, e2026290 118, https://doi.org/10.1073/pnas.2026290118, 2021.

175    Christensen, M. W., Jones, W. K., and Stier, P.: Aerosols enhance cloud lifetime and brightness along the stratus-to-cumulus transition, Proceedings of the National Academy of Sciences of the United States of America, 117, 17 591–17 598, https://doi.org/10.1073/pnas.1921231117, 2020.

Grise, K. M. and Kelleher, M. K.: Midlatitude Cloud Radiative Effect Sensitivity to Cloud Controlling Factors in Observations and Models: Relationship with Southern Hemisphere Jet Shifts and Climate Sensitivity, Journal of Climate, 34, 5869–5886,

180    https://doi.org/https://doi.org/10.1175/JCLI-D-20-0986.1, 2021.

Gryspeerdt, E., Quaas, J., and Bellouin, N.: Constraining the aerosol influence on cloud fraction, Journal of Geophysical Research, 121, 3566–3583, https://doi.org/10.1002/2015JD023744, 2016.

Hamby, D. M.: A review of techniques for parameter sensitivity analysis of environmental models, Environmental Monitoring and Assessment, 32, 135–154, https://doi.org/10.1007/BF00547132, 1994.

185    Li, W., Migliavacca, M., Forkel, M., Denissen, J. M. C., Reichstein, M., Yang, H., Duveiller, G., Weber, U., and Orth, R.: Widespread increasing vegetation sensitivity to soil moisture, Nature Communications, 13, 3959, https://doi.org/10.1038/s41467-022-31667-9, 2022.

Lundberg, S. M. and Lee, S. I.: A unified approach to interpreting model predictions, Advances in Neural Information Processing Systems, 2017-Decem, 4766–4775, 2017.

Lundberg, S. M., Erion, G. G., and Lee, S.-I.: Consistent Individualized Feature Attribution for Tree Ensembles, ArXiv, abs/1802.0, http:

190    //arxiv.org/abs/1802.03888, 2018.

Lundberg, S. M., Erion, G., Chen, H., DeGrave, A., Prutkin, J. M., Nair, B., Katz, R., Himmelfarb, J., Bansal, N., and Lee, S.-I.: From local explanations to global understanding with explainable AI for trees, Nature Machine Intelligence, 2, 56–67, https://doi.org/10.1038/s42256-019-0138-9, 2020.

Molnar, C.: Interpretable Machine Learning, 2 edn., https://christophm.github.io/interpretable-ml-book, 2022.

195    Myers, T. A., Scott, R. C., Zelinka, M. D., Klein, S. A., Norris, J. R., and Caldwell, P. M.: Observational constraints on low cloud feedback reduce uncertainty of climate sensitivity, Nature Climate Change, 11, 501–507, https://doi.org/10.1038/s41558-021-01039-0, 2021.

Scott, R. C., Myers, T. A., Norris, J. R., Zelinka, M. D., Klein, S. A., Sun, M., and Doelling, D. R.: Observed Sensitivity of Low-Cloud Radiative Effects to Meteorological Perturbations over the Global Oceans, Journal of Climate, 33, 7717–7734, https://doi.org/https://doi.org/10.1175/JCLI-D-19-1028.1, 2020.

200    Seddon, A. W. R., Macias-Fauria, M., Long, P. R., Benz, D., and Willis, K. J.: Sensitivity of global terrestrial ecosystems to climate variability, Nature, 531, 229–232, https://doi.org/10.1038/nature16986, 2016.

Wilson Kemsley, S., Ceppi, P., Andersen, H., Cermak, J., Stier, P., and Nowack, P.: A systematic evaluation of high-cloud controlling factors, EGUsphere, 2024, 1–32, https://doi.org/10.5194/egusphere-2024-226, 2024.

Yuan, T., Song, H., Wood, R., Oreopoulos, L., Platnick, S., Wang, C., Yu, H., Meyer, K., and Wilcox, E.: Observational evidence of strong

205    forcing from aerosol effect on low cloud coverage, Science Advances, 9, eadh7716, https://doi.org/10.1126/sciadv.adh7716, 2023.

---

## Author Response (AR4)

**Response to the reviewer #3: Analysis of the cloud fraction adjustment to aerosols and its dependence on meteorological controls using explainable machine learning      # EGUSPHERE-2023-1667**

Yichen Jia [1,2], Hendrik Andersen [1,2], and Jan Cermak [1,2]

[1]Karlsruhe Institute of Technology (KIT), Institute of Meteorology and Climate Research, Karlsruhe, Germany
[2]Karlsruhe Institute of Technology (KIT), Institute of Photogrammetry and Remote Sensing, Karlsruhe, Germany

**Correspondence:** Yichen Jia (yichen.jia@kit.edu)

We thank the third anonymous referee for the new round of review of the revised manuscript. Below, the reviewer's comments and suggestions are incorporated in italics and addressed hereafter, and the authors' responses are coloured in blue. Unless otherwise stated, line numbers in this document refer to the manuscript after the third-round review (before the updates following in this response letter).

5    **Referee 2**

**Specific comments**

1. *In this work, the authors use a novel method to identify the sensitivity of cloud fraction to aerosol variations. They find that Nd is strongly correlated to CF, after accounting for the impact of other cloud controlling factors. This work is clearly in scope for ACP and would be of interest to its readers. The work is of a good standard and I think would be*
10    *suitable for publication with a few additions/changes.*

   *I appreciate the authors have already done a considerable amount of work responding to other reviewers, I would only have a few small points to add here.*

   Thank you for your positive evaluation of our manuscript. We have addressed your specific comments individually below.

2. *It seems that the interpretation of the SHAP values are not straightforward. This is not a fault of the author or the reader,*
15    *but given this method is still fairly new, a little extra explanation could be useful. I am not sure directing the reader to read a textbook targeted at computer scientists is useful - with a small amount of extra text, this paper could provide an explanation of the SHAP-based techniques accessible at a broader range of atmospheric scientists and help encourage the use of this technique.*

   *As I undersatnd it, the SHAP value does not represent a sensitivity (as the authors say),. However, this was not immedi-*
20    *ately clear to me (having not looked at this in much detail before). Phrases like 'way to measure the relative contributions of Nd (as a surrogate for aerosols) and meteorological factors to CLF changes' initially suggested to me that we were looking at a sensitivity-like measure. Assuming I am understanding this correctly, the 'base value' here is effectively the*

*climatological CLF at a given location (what would be predicted with no extra information. The SHAP values of Nd then*
*show the CLF that would be predicted, given that the Nd is known (or at least the difference from the climatological*
*CLF), such that $CLF|Nd = SHAP(Nd) * sigma_C LF + CLF_c lim$. To me, this framing then makes it clearer that a*
*more traditional sensitivity (dCF/dlnNd) would have to be calculated using dSHAP(Nd)/dlnNd and CLF scaling.*

Thank you for your insightful comments. We agree that providing a clearer explanation of this method will enhance the comprehensibility of the manuscript within the field of atmospheric sciences. Thus, we have rephrased Sect. 2.3.1 as follows:

(a) We agree the sentence the reviewer mentioned at lines 179 and 180 "It provides a novel ... to CLF changes" might lead to an incorrect first impression of SHAP values. Therefore, we have removed it.

(b) We have rephrased the sentence "The contribution of ... besides global feature importance "from line 182 to line 185 to distinguish between local and global explanations: "The contribution of a predictor value to a specific model prediction is calculated as the difference between the predictions of the model in the presence and absence of this particular predictor for all possible combinations of predictor values. Since this is performed at a "local" level (i.e. for this specific instance's prediction), it allows for insights into how a certain model outcome is achieved, thereby complementing more traditional "global" (considering all instances) feature importance measures (e.g. partial dependence plot)."

(c) Line 186: We have included an explanation of the base value "what would be predicted in the absence of any feature information".

(d) Line 190: We find explaining the base value as the climatological CLF to be appropriate and inspiring, and it helps the atmospheric science community more quickly understand the SHAP approach. We have incorporated it into the manuscript: "The base value could be analogous to the climatological CLF for a given geographical window assuming no information about the input parameters is known. In this context, the SHAP values of input features indicate the extent to which knowing information about each feature value would deviate the prediction from the climatological CLF (base value)." The structure of this section has also been altered accordingly.

3. *I am a little concerned by the very linear relationship in log space shown in Fig. 1b. Given CLF is capped at 0 and 100 %, many previous studies have found a non-linear relationship even in lnNd-space (e.g. Gryspeerdt et al, JGR, 2016). It is not clear to me how such a linear relationship here can be achieved, especially when using instantaneous daily cloud property data?*

Thank you for highlighting this important point for discussion. In fact, the very linear relationship shown in Fig. 1 (b) does not hold for all $5° \times 5°$ grid cells. Here, we showcase some additional SHAP dependence plots where the points on the scatter plot do not form a distinctly linear relationship (see Fig. 1 in this document). Naturally, if we plot $N_d$ without taking the logarithm, the nonlinearity will be much more apparent. The reason appears to be that, in certain $5° \times 5°$ grid

boxes, the XGB + SHAP framework is unable to capture the nonlinearity of the relationship, despite the models being tuned for daily data at these specific grid boxes.

It seems that there is no relationship between CLF and $\ln N_d$ explicitly reported in log space in Gryspeerdt et al. (2016) that we can directly compare with. They only presented CLF–AOD relationships, mediated by $N_d$, in log space for their global patterns. However, it appears that their regional CLF–AOD, $N_d$–AOD and CLF–$N_d$ relationships using joint histograms and probability distributions are not based on log-transformed $N_d$ and AOD. We are unsure if the reviewer precisely referred to the nonlinear regional relationships by conditional probabilities and joint histograms, or the lnAOD–$N_d$–CLF sensitivity using linear regression in this study.

That being said, generally, the nonlinear relationships in Gryspeerdt et al. (2016) would suggest that the nonlinearity in this system might be better captured using joint histograms and conditional probabilities in the regions where the relationship is notably linear in our study. This makes sense because their method is designed to retain more information about the nonlinearity and report nonlinear relationships across different regions and cloud regimes. It is worth mentioning that the relationships in Gryspeerdt et al. (2016) based on conditional probability distributions were analysed for three $20° \times 20°$ regions and one $25° \times 25°$ region, which are larger than our $5° \times 5°$ geographical windows. We could expect similarly nonlinear relationships by training XGB models for the same four regions. This is an interesting comparison for our future work.

After presenting these regional results, Gryspeerdt et al. (2016) subsequently showed the spatial patterns of the sensitivities as we do in our study. The sensitivity values on their maps are also calculated as the slope of linear regressions, meaning that some nonlinear and "convolved" relationships are disregarded as well. The main goal of our study is also to show the spatial patterns of sensitivity. The philosophy of our study is to use linear regression to capture as much of the relationship revealed by the explainable machine learning framework as possible. Therefore, in the grid boxes where the XGB models may not fully capture the nonlinearity, we can at least be confident that using simple linear regression reduces the loss of information from the XGB model.

We have now summarised and incorporated the discussion into Sect. 2.3.2 in the manuscript. Figure 1 in this document has also been included in the supplementary material as Fig. S1. The numbering of the figures in the supplementary material has been updated accordingly:

"It should be noted that the notably linear relationship in Fig. 1 (b) does not hold across all geographical windows. Fig. S1 displays additional exemplary windows where the relationships exhibit less linearity. Our approach also captures non-linearity in the system; in these cases, the linear regression helps decrease the convolved relationships as in Gryspeerdt et al. (2016)."

[Figure]

**Figure 1.** SHAP dependence plots similar to Fig. 1 (b) illustrating relatively nonlinear relationships between $\ln N_d$ SHAP values and feature values (standardized). The latitude and longitude values for each subplot represent the midpoint of each $5° \times 5°$ geographical window.

85   4. *There are good arguments for using the normalised values, particularly when comparing the different controls against each other. However, it is also important to be able to compare the results of this work against other studies, where the use of normalised values are less common. It would be good to have either a beta value that can be compared to the range in Bellouin et al., Rev. Geophys. (2020), or a forcing estimate that could be compared to those from other studies (e.g. Gryspeerdt et al, ACP, 2020).*

90   Thank you very much for your suggestions. We agree that there are studies where the CLF response to aerosol (proxies) is not directly compared with the quantification of the effects of meteorological cloud-controlling factors. Comparison with these studies will improve the scientific significance of this manuscript.

In the supplementary material, Fig. S4 shows the spatial patterns of the CLF-$N_d$ sensitivity without standardization. In other words, the global average of these sensitivity values is comparable to the $\beta$ value in Bellouin et al. (2020). We have
95   added this comparison in Sect. 3.2.2 of the manuscript:

"The global weighted average of the CLF-ln$N_d$ sensitivity without standardization is 0.112 (unitless), and its spatial pattern is shown in Fig. S4. This value is higher than the upper bound of 0.1 reported by Bellouin et al. (2020), which is based on global climate models and large eddy simulations. This may be partly due to the aforementioned bias. However, it is important to note that our non-standardized CLF-$N_d$ sensitivity, shown in Fig. 1 (a), closely mirrors that from
100   Yuan et al. (2023) with a similar range. In addition, the high lnCLF–ln$N_d$ values estimated in Chen et al. (2022, 2024) suggest that values exceeding the upper bound of 0.1 might be plausible. These recent observational studies, including quantifying cloud fraction adjustment based on ship tracks Yuan et al. (2023), volcano aerosol perturbations (Chen et al., 2022, 2024), and our SHAP approach using global satellite observations, indicate that the 0.1 upper bound may be extended. In future work, estimating a radiative forcing using the SHAP-based sensitivities will make our study more
105   comparable with other research on cloud fraction adjustment.

5. *I also found the mention/discussion of the potential non-causal Nd-CF link to be a bit lacking, especially around the headline results in the abstract and conclusions. While it is mentioned that there are potential retrieval biases in Nd as a function of CF, the impact of these is somewhat downplayed in the interpretation of the results. There are significant uncertainties in the retrieval of Nd in broken-cloud regimes, exactly where the observed Nd-CF relationship is strongest.*
110   *While these results are clearly consistent with the theoretical behaviour of stratocumulus clouds, statements that CLF is 'sensitive to' Nd and that aerosol has a 'considerable impact on MBL cloudiness' might be over-attributing the causality of the results. This doesn't require a large change in the paper, just a little bit of rewording of some of the results.*

Thank you for your feedback. Although we have included a separate section to discuss the limitations, including the non-causal $N_d$–CLF relationship, and have rephrased sentences throughout the manuscript, we acknowledge that some
115   statements may still be overly assertive. Therefore, we have revised the manuscript to provide a more cautious interpretation of the results, including updates to the abstract, method, conclusion, and results sections:

(a) Abstract, lines 10–11: We have revised the summary of the results concerning the $N_d$–CLF sensitivity: "Based on our statistical approach, global patterns of CLF sensitivity suggest that CLF is positively associated with $N_d$, particularly in the stratocumulus-to-cumulus transition regions and the southern hemispheric midlatitudes. However, $N_d$ retrieval bias may contribute to non-causality in these positive sensitivities, and hence they should be considered as upper-bound estimates."

(b) Sect. 2.3.3, line 258: We have added "For example, the subpixel effect can introduce more bias in the $N_d$ retrieval process within broken-cloud regimes due to increased heterogeneity. The $N_d$ retrieval biases are ..."

(c) Sect. 3.2.2, line 342: "CLF is particularly sensitive to Nd in the regions of ..." has been reworded as "The relationship between CLF and $N_d$ is found particularly strong in the regions of ..."

(d) Line 346: We have updated the sentence as: "However, as this cloud regime transition involves clouds shifting from more overcast to more broken, the strong relationships in these regions may be more affected by $N_d$ retrieval errors."

(e) Sect. 4 Conclusions, line 466: The last sentence of this paragraph has been revised as: "The main findings of this study, which should be interpreted in light of the data and methodology limitations discussed in Sect. 2.3.3, are summarized as follows:"

(f) Sect. 4 Conclusions, point 1, lines 470 to 472: reworded as "The estimated $N_d$–CLF sensitivity and its magnitude suggest that aerosols likely have a considerable impact on MBL cloudiness, although this may partially result from an overestimation caused by the effect of a positive retrieval bias of $N_d$ at high CLF."

**Other minor suggestions**

- *Is the temporal split of train-test data suitable, given the underlying change in climate? I doubt it would affect the results much, but could be something to consider for future work.*

Thank you for bringing this up, we think this is a valuable point for discussion. We acknowledge that even within the relatively short period we studied (2011-2019), climate data from different periods may exhibit differences in data characteristics due to factors such as year-to-year variations and extreme weather events. We agree with the reviewer that different methods of partitioning data into training and test sets can influence the model, which is an interesting point for future investigations. Besides the simple chronological non-random split, strategies like the one proposed by Salazar et al. (2022), which handles spatial autocorrelation, could also be implemented in future work.

The temporal train-test split has been commonly applied in many studies using machine learning to analyze aerosol-cloud-climate interactions (e.g. Fuchs et al., 2018; Dadashazar et al., 2021; Bender et al., 2024; Chen et al., 2024). However, some of these studies split the data sets by random shuffling, which may not be the best practice.

In Earth sciences, data tend to be structured spatiotemporally and may have dependencies on nearby data points (Karpatne et al., 2017). A random train-test split can lead to temporally autocorrelated training and testing data from neighbouring

time steps, resulting in overoptimistic model performance evaluations (Roberts et al., 2017; Beucler et al., 2023). In addition, this random partitioning breaks the natural sequential order and results in data leakage, as the model already has information from the future while being trained (Malik, 2020; Kapoor et al., 2023). Since this "time travelling" does not align with real-world prediction scenarios, the test score cannot accurately reflect how well the model generalizes to unseen data. Therefore, the chronological temporal train-test split in our study, as done in Andersen et al. (e.g. 2023), is preferable to a random split for more realistic and reliable model performance. Furthermore, the 5-fold cross-validation used for hyperparameter tuning in our study is also done non-randomly to avoid this problem.

In Fig. 1, we examined how different training-test splits (80/20 %, 75/25 %, 50/50 %) affect the sensitivity and IAI for the exemplary window shown in Fig. 1 of the manuscript. Although we were unable to assess the global impact of different split ratios on the final sensitivities and interaction effects due to limitations in time and computational resources, our results from the example window indicate that varying the split ratios only slightly changes model performance ($R^2$). Importantly, the final results based on the SHAP approach remain consistent.

Accordingly, we have revised the sentence at lines 158-159 as "By chronologically splitting the training and test sets without random shuffling, we ensure that the training data will not see future information and the autocorrelation in data will not lead to overopstimic evaluation of the model's performance Beucler et al. (2023); Kapoor et al. (2023).

- *Are the Nd values calculated from the 1x1 degree tau-re values, or from the underlying joint histograms? The use of the large-scale mean values may lead to biases, due to the non-linearity of the Nd calculation. If it is useful, there is pre-filtered Nd data available for the period of study as a companion to Gryspeerdt et al., ACP, 2022.*

Thank you for raising this point. The $N_d$ values are calculated from the $1° \times 1° r_e$ and $\tau_c$ values. We have acknowledged a caveat in the manuscript that using large-scale mean values in our study may bias our results. The existing $N_d$ data sets by Gryspeerdt et al. (2022) appear to be good for our use. However, in addition to filtering criteria for solar zenith and satellite viewing angles, the "G18" sampling strategy also filters out pixels with a 5 km CLF smaller than 0.9 for more homogeneous clouds. Since we aim to analyse the relationship between CLF and the predictors, including only CLF values larger than 0.9 may not be appropriate because it would limit our analysis to a small portion of these relationships. In future work, $N_d$ calculated based on underlying joint histograms could be a better choice. Exploring the impact of using mean values versus joint histograms on our quantification of sensitivities and interaction effects is another interesting point for the follow-up research.

We have added into Sect. 2.3.3 Data limitation in the manuscript: "Another caveat in our data is that $N_d$ values in our study are computed using MODIS level-3 large-scale mean $r_e$ and $\tau_c$ values instead of joint histograms as in Gryspeerdt et al. (2016). This may introduce additional biases considering the nonlinearity of the $N_d$ calculation. In future work, $N_d$ data calculated from underlying joint histograms or pre-filtered data by Gryspeerdt et al. (2022) could be applied to be compared with the results in this study."

[Figure]

(a) Test size = 20 %, $R^2 = 0.49$

(b) Test size = 25 %, $R^2 = 0.50$

(c) Test size = 50 %, $R^2 = 0.44$

**Figure 2.** Examples illustrate that varying train-test split ratios affects only the test score $R^2$, but does not alter the quantification of sensitivities.

**Minor modifications independent of the reviewer comments**

Line 161: "As suggested by Karpatne et al. (2017), a single ML model may not perform well across all regions due to the heterogeneity of relevant processes. Therefore, data ..."

Line 161: "regionally-specific relationships" to "regional relationships".

Line 187: "data points" to "data set"

Line 228: "Figure. S1" to "Fig. S2"

Line 339: "the" has been inserted between "of" and "MBLC".

Line 348: "AOD" to "aerosol optical depth"

---

## Author Response (AR5)

**Response to the Reviewer #3: Analysis of the cloud fraction adjustment to aerosols and its dependence on meteorological controls using explainable machine learning # EGUSPHERE-2023-1667**

Yichen Jia [1,2], Hendrik Andersen [1,2], and Jan Cermak [1,2]

[1]Karlsruhe Institute of Technology (KIT), Institute of Meteorology and Climate Research, Karlsruhe, Germany
[2]Karlsruhe Institute of Technology (KIT), Institute of Photogrammetry and Remote Sensing, Karlsruhe, Germany

**Correspondence:** Yichen Jia (yichen.jia@kit.edu)

We thank the Referee #3 for the review of the revised manuscript. Below, the reviewer's comments and suggestions are incorporated in italics and addressed hereafter, and the authors' responses are coloured in blue. Unless otherwise stated, line numbers in this document refer to the manuscript after the third-round review (before the updates following in this response letter).

**Referee 2**

**Specific comments**

1. *I thank the authors for their changes to the manuscript and I look forward to the final version of the paper.*

   Thank you for your positive feedback on our revised version.

2. *I would note that although the authors state that they can't use the G19 Nd dataset from the Gryspeerdt et al (2022) output, if they are currently performing their filtering on a 1x1 degree grid - they could apply their filtering to the Q06 Nd dataset instead to achieve the 'non-CLF filtered' Nd dataset they require. This is a minor point (and I appreciate that this bias is likely small and would require significant additional work to fix). I might instead recommend that the authors note (perhaps where they discuss the potential impact of Nd biases) that small-scale sampling issues can have large impacts on estimates of aerosol-cloud relationships (e.g. Arola et al, 2022 - 10.1038/s41467-022-34948-5)*

   Thank you for your insightful feedback. We appreciate your recognition that this point is minor, and we thank you for acknowledging that the potential bias is likely small and that implementing the change would require significant additional effort. We also value your suggestion on the possible use of the Q06 $N_d$ data set. We recognize that applying additional filtering for solar zenith and satellite zenith to the Q06 data set could provide a viable alternative, allowing us to retain the full CLF value range. As we already noted in the revised manuscript (Line 275), we plan to implement the Gryspeerdt et al. (2022) data set in our companion study, where we aim to compare observational results with those from the ICON-HAM climate model using the same machine learning framework.

   We agree that it is a good idea and a more practical approach to incorporate the impact of small-scale sampling issues on estimating aerosol-cloud relationships. In response to the reviewer's suggestion, we have included the following

discussion in Sect. 2.1, Line 114: "Furthermore, the interpretation of the causal effect of $N_d$ on CLF can also be obscured by small-scale sampling issues. In particular, apart from the retrieval errors in $r_e$ and $\tau_c$, the natural spatial variability in cloud fields can also propagate into $N_d$ estimates and distort the $N_d$–CLF relationship (Arola et al., 2022; Liu et al., 2024)."

3. *The only other technical point I would make is that the text on Fig. 1 is quite small. This might be corrected in typesetting, but I wanted to note it just incase.*

Thank you for pointing this out. We have already adjusted Fig. 1 as well as the corresponding non-standardized plots (Fig. S2).

**Minor modifications independent of the reviewer comments**

Line 106: "This retrieval approach relies..." has been modified to "This derivation approach relies..."

**References**

35    Arola, A., Lipponen, A., Kolmonen, P., Virtanen, T. H., Bellouin, N., Grosvenor, D. P., Gryspeerdt, E., Quaas, J., and Kokkola, H.: Aerosol effects on clouds are concealed by natural cloud heterogeneity and satellite retrieval errors, Nature Communications, 13, 7357, https://doi.org/10.1038/s41467-022-34948-5, 2022.

     Gryspeerdt, E., Mccoy, D. T., Crosbie, E., Moore, R. H., Nott, G. J., Painemal, D., Small-griswold, J., Sorooshian, A., and Ziemba, L.: The impact of sampling strategy on the cloud droplet number concentration estimated from satellite data, pp. 3875–3892, 2022.

40    Liu, Y., Lin, T., Zhang, J., Wang, F., Huang, Y., Wu, X., Ye, H., Zhang, G., Cao, X., and de Leeuw, G.: Opposite effects of aerosols and meteorological parameters on warm clouds in two contrasting regions over eastern China, Atmospheric Chemistry and Physics, 24, 4651–4673, https://doi.org/10.5194/acp-24-4651-2024, 2024.